# A radio-detected type Ia supernova with helium-rich circumstellar material

Erik C. Kool[1 ✉], Joel Johansson[1,2], Jesper Sollerman[1], Javier Moldón[3,4], Takashi J. Moriya[5,6], Seppo Mattila[7,8], Steve Schulze[2], Laura Chomiuk[9], Miguel Pérez-Torres[3,10], Chelsea Harris[9], Peter Lundqvist[1], Matthew Graham[11], Sheng Yang[1,12], Daniel A. Perley[13], Nora Linn Strotjohann[14], Christoffer Fremling[11], Avishay Gal-Yam[14], Jeremy Lezmy[15], Kate Maguire[16], Conor Omand[1], Mathew Smith[15,17], Igor Andreoni[18,19,20], Eric C. Bellm[21], Joshua S. Bloom[22,23], Kishalay De[24], Steven L. Groom[25], Mansi M. Kasliwal[11], Frank J. Masci[25], Michael S. Medford[22,23], Sungmin Park[26], Josiah Purdum[27], Thomas M. Reynolds[28], Reed Riddle[11], Estelle Robert[15], Stuart D. Ryder[29,30], Yashvi Sharma[11] & Daniel Stern[31]

Type Ia supernovae (SNe Ia) are thermonuclear explosions of degenerate white dwarf stars destabilized by mass accretion from a companion star[1], but the nature of their progenitors remains poorly understood. A way to discriminate between progenitor systems is through radio observations; a non-degenerate companion star is expected to lose material through winds[2] or binary interaction[3] before explosion, and the supernova ejecta crashing into this nearby circumstellar material should result in radio synchrotron emission. However, despite extensive efforts, no type Ia supernova (SN Ia) has ever been detected at radio wavelengths, which suggests a clean environment and a companion star that is itself a degenerate white dwarf star[4,5]. Here we report on the study of SN 2020eyj, a SN Ia showing helium-rich circumstellar material, as demonstrated by its spectral features, infrared emission and, for the first time in a SN Ia to our knowledge, a radio counterpart. On the basis of our modelling, we conclude that the circumstellar material probably originates from a single-degenerate binary system in which a white dwarf accretes material from a helium donor star, an often proposed formation channel for SNe Ia (refs. 6,7). We describe how comprehensive radio follow-up of SN 2020eyj-like SNe Ia can improve the constraints on their progenitor systems.

SN 2020eyj was first detected on 7 March 2020 UT (modified Julian date (MJD) = 58,915.12; see 'Observations' section in Methods), at $\alpha = 11$ h 11 min 47.19 s, $\delta = 29°$ 23′ 06.5″ (J2000). The SN was classified as a SN Ia (ref. 8) based on a low-resolution spectrum obtained on 2 April 2020, 25 days after the first detection. Comparisons with type Ia and Ibc spectra from the literature support the SN Ia classification (see 'SN Ia classification' section in Methods and Fig. 1). Unusual evolution of the later light curve prompted us to obtain a second spectrum on 20 July 2020, 131 days after first detection. The second spectrum

was very similar to those of type Ibn SNe (SNe Ibn), which are SNe that interact with helium-rich circumstellar material (CSM) and have spectra characterized by narrow (roughly a few $10^3$ km s⁻¹) He I emission lines while showing little to no H I (refs. 9,10).

On the basis of the late-time (tail-phase) CSM-interaction-dominated spectra (Fig. 2), SN 2020eyj falls in the category of the rare subclass of SNe Ia that show evidence of CSM interaction in their optical spectra (SNe Ia–CSM; ref. 11). The narrow emission lines in the spectra of such interacting SNe arise from shock interaction between the fast-moving

[1]The Oskar Klein Centre, Department of Astronomy, Stockholm University, AlbaNova, Stockholm, Sweden. [2]The Oskar Klein Centre, Department of Physics, Stockholm University, AlbaNova, Stockholm, Sweden. [3]Instituto de Astrofísica de Andalucía, Consejo Superior de Investigaciones Científicas (CSIC), Granada, Spain. [4]Jodrell Bank Centre for Astrophysics, School of Physics and Astronomy, The University of Manchester, Manchester, UK. [5]National Astronomical Observatory of Japan, National Institutes of Natural Sciences, Mitaka, Japan. [6]School of Physics and Astronomy, Faculty of Science, Monash University, Clayton, Victoria, Australia. [7]Tuorla Observatory, Department of Physics and Astronomy, University of Turku, Turku, Finland. [8]School of Sciences, European University Cyprus, Nicosia, Cyprus. [9]Center for Data Intensive and Time Domain Astronomy, Department of Physics and Astronomy, Michigan State University, East Lansing, MI, USA. [10]Facultad de Ciencias, Universidad de Zaragoza, Zaragoza, Spain. [11]Division of Physics, Mathematics and Astronomy, California Institute of Technology, Pasadena, CA, USA. [12]Henan Academy of Sciences, Zhengzhou, China. [13]Astrophysics Research Institute, Liverpool John Moores University, Liverpool, UK. [14]Department of Particle Physics and Astrophysics, Weizmann Institute of Science, Rehovot, Israel. [15]Univ. Lyon, Univ. Claude Bernard Lyon 1, CNRS/IN2P3, IP2I Lyon, UMR 5822, Villeurbanne, France. [16]School of Physics, Trinity College Dublin, The University of Dublin, Dublin, Ireland. [17]School of Physics and Astronomy, University of Southampton, Southampton, UK. [18]Joint Space-Science Institute, University of Maryland, College Park, MD, USA. [19]Department of Astronomy, University of Maryland, College Park, MD, USA. [20]Astrophysics Science Division, NASA Goddard Space Flight Center, Greenbelt, MD, USA. [21]DIRAC Institute, Department of Astronomy, University of Washington, Seattle, WA, USA. [22]Department of Astronomy, University of California, Berkeley, Berkeley, CA, USA. [23]Lawrence Berkeley National Laboratory, Berkeley, CA, USA. [24]Kavli Institute for Astrophysics and Space Research, Massachusetts Institute of Technology, Cambridge, MA, USA. [25]Infrared Processing and Analysis Center (IPAC), California Institute of Technology, Pasadena, CA, USA. [26]Ulsan National Institute of Science and Technology, Ulsan, South Korea. [27]Caltech Optical Observatories, California Institute of Technology, Pasadena, CA, USA. [28]The Cosmic Dawn Center (DAWN), Niels Bohr Institute, University of Copenhagen, Copenhagen, Denmark. [29]School of Mathematical and Physical Sciences, Macquarie University, Sydney, New South Wales, Australia. [30]Astronomy, Astrophysics and Astrophotonics Research Centre, Macquarie University, Sydney, New South Wales, Australia. [31]Jet Propulsion Laboratory, California Institute of Technology, Pasadena, CA, USA. ✉e-mail: erik.kool@astro.su.se

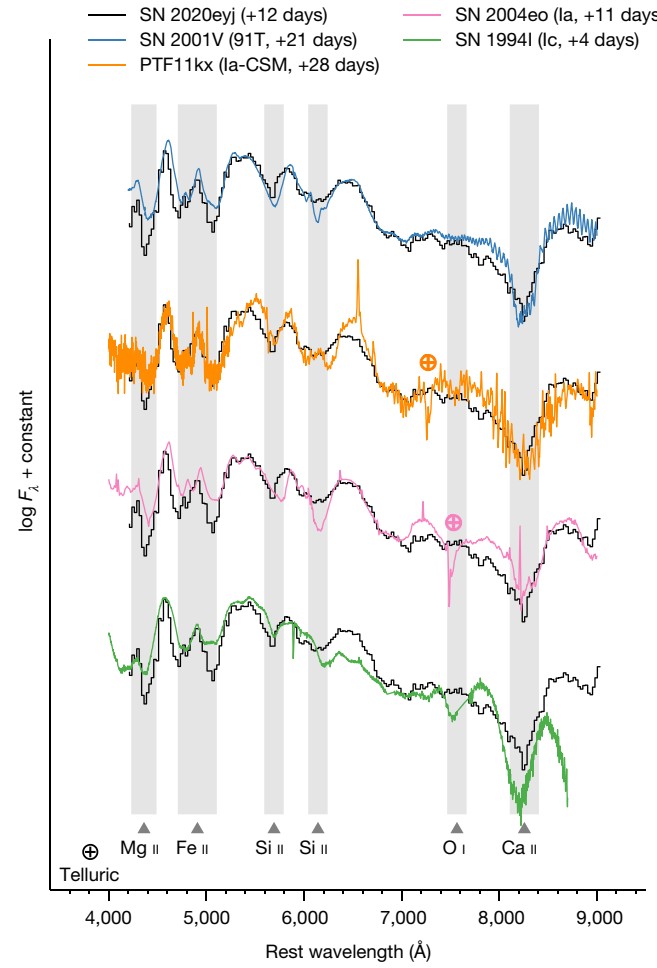

**Fig. 1 | The first spectrum of SN 2020eyj is consistent with a type Ia(−CSM).** The SEDM classification spectrum of SN 2020eyj, obtained about 12 days after peak and shown in black, is compared with type Ia-91T SN 2001V, type Ia–CSM PTF11kx, type Ia SN 2004eo and type Ic SN 1994I. Phases are relative to peak, which—in the case of SN 2020eyj—has an uncertainty of a couple of days. Several important absorption features are indicated at the expected wavelengths. Notably, the spectrum of SN 2020eyj lacks any sign of O I 7,774 Å absorption. Spectra have been corrected for MW reddening. Telluric features are indicated by crossed circles.

SN ejecta and the slow-moving CSM[12]. SNe Ia–CSM are strong contenders for the single-degenerate (SD) SN Ia formation channel on account of the CSM, which is commonly assumed to originate from a non-degenerate donor star through stellar or accretion winds. Before SN 2020eyj, all of the discovered SNe Ia–CSM exhibited prominent Balmer emission lines and only weak He emission features[11].

Typically, CSM interaction contributes substantially to or even dominates the spectral and light-curve evolution of SNe Ia–CSM from the beginning, hindering unambiguous classification as SNe Ia (ref. 13). However, in some rare cases, SNe Ia–CSM have shown a delay in CSM interaction[14–16], suggesting that the CSM was located far (>$10^{15}$ cm) from the binary system at the time of explosion. Notably, PTF11kx cemented SNe Ia–CSM as a bona fide SN Ia subclass by virtue of a delay of about 60 days, allowing for an indisputable SN Ia classification before CSM interaction[15]. SN 2020eyj follows a similar evolution as PTF11kx, initially showing a typical SN Ia bell-shaped light curve (Fig. 3) and a spectrum consistent with a SN Ia of the 91T subgroup[17] without clear evidence for CSM interaction (Fig.1). Then, at 50 days after first detection, the g-band light curve of SN 2020eyj diverges from a steady decline into a plateau that lasts for roughly 200 days. Such an evolution and colour change is

not expected for a normal SN Ia (Fig. 3) but is driven by the emergence of spectral features associated with CSM interaction (see 'Light-curve analysis' section in Methods). We interpret the start of the plateau at 50 days as the epoch when CSM interaction starts to contribute substantially or even dominate the light curve of SN 2020eyj. Assuming a SN ejecta velocity of $10^4$ km s$^{-1}$ (ref. 18), the delay corresponds to an inner boundary to the CSM of about $4 × 10^{15}$ cm. Except for the presence of He emission lines, the late-time spectra of SN 2020eyj are typical for the SN Ia–CSM class, with prominent broad Ca II emission from the near-infrared (NIR) triplet and without any sign of O I$λ$7774 emission (Fig. 2). The compact and star-forming host galaxy of SN 2020eyj (see 'Host galaxy' section in Methods) is also consistent with those of other SNe Ia–CSM[11].

Despite the similarities between SN 2020eyj and other SNe Ia–CSM, the presence of He I lines and absence of prominent H I lines remains a striking difference with profound implications for the progenitor system. As H I is easier to ionize than He I, the absence of the lines indicates that the CSM around SN 2020eyj, and thus the companion star, is He-rich and H-poor. Although the late-time spectra of SN 2020eyj are similar to those of SNe Ibn, these SNe are presumed to arise from the core collapse of massive (>10 $M_{\odot}$) stars[9,19,20], which are unlikely to be in a binary system with a white dwarf (WD), as they would undergo core collapse long before the WD formed. A merger involving a degenerate He WD donor star is also disfavoured, because in such merger models, only a small amount of unburned He (about 0.03 $M_{\odot}$ (ref. 21)) is present close to ($\lesssim 10^{12}$ cm) the WD (ref. 22), whereas the CSM around SN 2020eyj resides at >$10^{15}$ cm. Instead, a strong candidate for the donor star in the SN 2020eyj progenitor system is a non-degenerate He star (initial mass 1–2 $M_{\odot}$, for example, ref. 23). WD + He star systems can be formed by means of binary evolution[24] and this SD channel for SNe Ia has garnered recent interest because the very restrictive limits placed by radio non-detections and deep optical imaging[25] that exclude most H-rich donor star models still allow for low-CSM-density WD + He star systems[25,26]. The possible detection in pre-explosion Hubble Space Telescope imaging of the progenitor system of the type Iax (SNe Ia similar to SN 2002cx (ref. 27)) SN 2012Z, a blue compact source interpreted as a He-star donor[28], has further strengthened this hypothesis, although the thermonuclear nature of type Iax SNe is debated[29].

The CSM interaction in SN 2020eyj is also confirmed, for the first time in a SN Ia, through the detection of a radio counterpart, at a frequency of 5.1 GHz at 605 and 741 days after the first detection (see 'Radio' section in Methods). Follow-up in the X-rays did not yield a detection (see 'X-ray' section in Methods). We model the radio synchrotron emission, which results from the shock interaction between the ejecta and the CSM, assuming two basic CSM configurations expected in a SD progenitor system; a constant density shell and a wind-like density profile with density $ρ ∝ r^{-2}$ (Fig. 4). A constant density shell could result from a mass ejection event such as a nova, whereas a wind-like CSM profile would be expected from an optically thick wind, in which the mass-transfer rate from the donor star to the WD exceeds the maximum accretion rate of He-rich material that the WD can burn on its surface[26,30]. As well as CSM material resulting from a SD scenario, we consider synchrotron emission resulting from the interaction of a SN Ia from a double-degenerate (DD) WD merger interacting with the local interstellar medium (ISM)[31]. For the SD shell model, the radio detections are best explained with a CSM mass of $M_{csm} = 0.36$ $M_{\odot}$ (see 'CSM shells' section in Methods), with the expectation that the radio light curve will start to drop off rapidly at around 900 days. For the SD optically thick wind model, the bolometric light-curve tail (see 'Bolometric light curve' section in Methods) and radio detections of SN 2020eyj are well fitted with a mass-transfer rate of $10^{-3}$–$10^{-2}$ $M_{\odot}$ yr$^{-1}$, microphysics parameter $\epsilon_B = 10^{-5}$–$10^{-3}$ and a CSM mass within $10^{17}$ cm of $M_{csm} = 0.3$–1.0 $M_{\odot}$. The DD ISM model (the dashed lines in Fig. 4) requires unusually high ISM densities and does not recover the observed decline in flux, ruling out the DD formation channel for SN 2020eyj (see 'ISM' section in Methods). The best fit radio light curves of

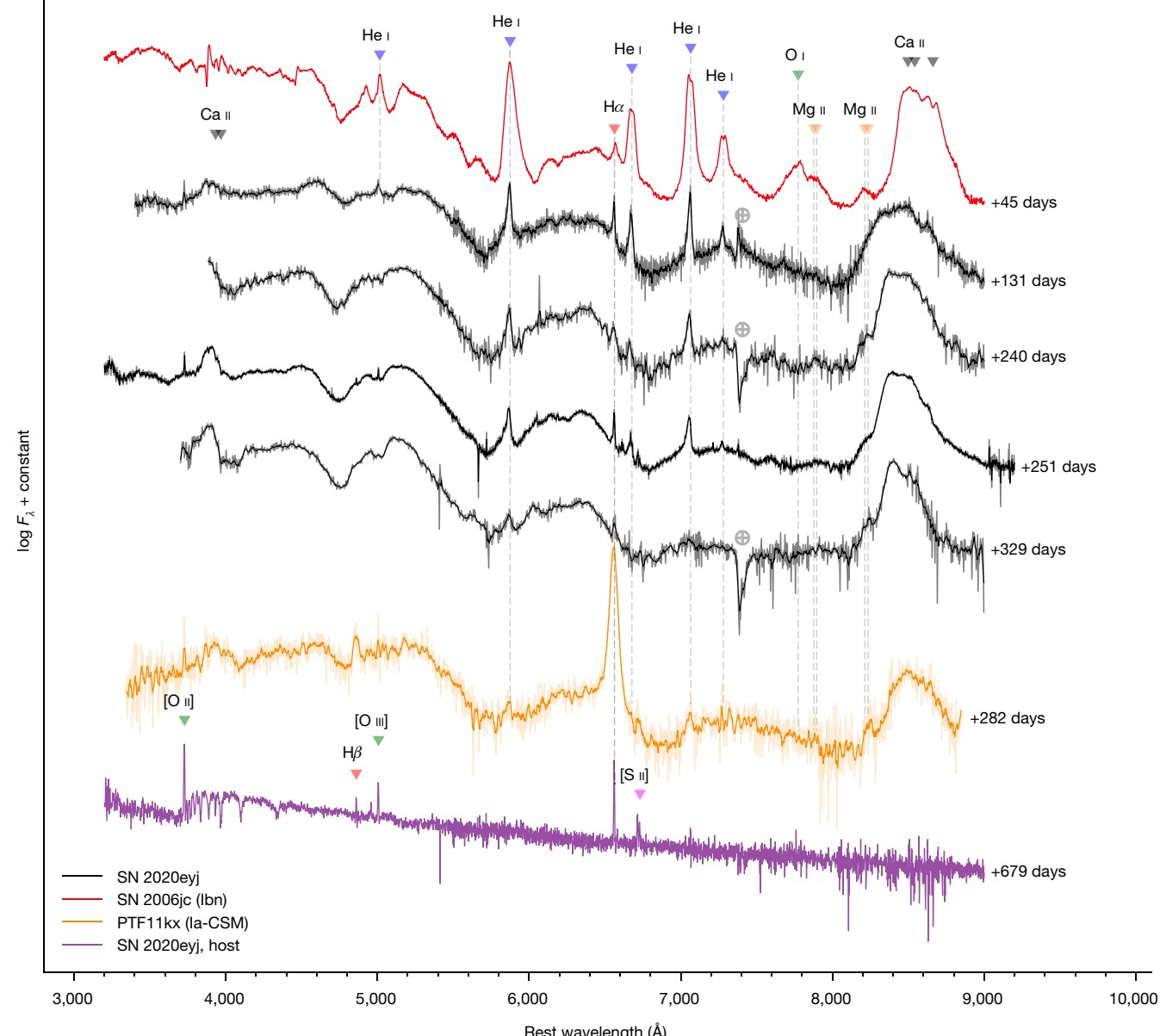

**Fig. 2 | The spectra of SN 2020eyj in the tail phase are dominated by CSM interaction.** The spectra of SN 2020eyj at late phases (in black) are compared with the prototypical type Ibn SN 2006jc and the type Ia–CSM SN PTF11kx. The spectra show features common to SNe Ia–CSM, such as the quasi-continuum blueward of 5,700 Å and broad Ca II emission. The main SN emission features are identified in the top spectrum. The emission lines in SN 2020eyj show strong asymmetry, with attenuated red wings (Extended Data Fig. 3). The bottom spectrum is of the host of SN 2020eyj, obtained at 679 days, some 300 days after the SN had faded below the detection limit of the ZTF. Some unresolved galaxy lines are marked. Phases are relative to first detection, which—in the case of SN 2006jc—was at or after the peak. Spectra have been corrected for MW reddening. Telluric features are indicated by crossed circles.

the shell and wind models differ in particular at early phases (Fig. 4), but no radio data were obtained at these epochs. Instead, multifrequency monitoring of the radio counterpart of SN 2020eyj until late phases (>1,000 days) will allow to discriminate between the rapid drop-off of the shell model and a shallower decline expected in the case of a wind-like CSM.

A viable progenitor scenario for SN 2020eyj needs to explain not only the presence and properties of a He-rich CSM but also its detached configuration. For the delayed type Ia–CSM SN 2002ic, the CSM-free cavity was attributed to a possible drop-off in mass-transfer rate or the emergence of a low-density fast wind evacuating the CSM[32]. In the case of PTF11kx, the delayed CSM interaction was explained by a scenario involving a symbiotic nova progenitor, in which recurrent novae on the surface of the WD sweep up the wind-deposited CSM into shells[15]. SN 2020eyj shows strong similarities to PTF11kx, which may hint at a

common progenitor scenario. Their light curves are virtually identical up until day 50 (Extended Data Fig. 1) with risetimes of about 14 days in the *g* band, which is fast for a SN Ia (ref. 33). And, except for the nature of the narrow emission lines, both SNe have similar spectra throughout their evolution (Figs. 1 and 2). For SN 2020eyj, a nova progenitor could look like V445 Puppis (V445 Pup; see 'V445 Puppis' section in Methods), the only known nova system that showed He-rich, but H-free, ejecta[34,35]. Notably, the V445 Pup system is considered a prime candidate progenitor system for the He star + WD SN Ia channel, as it is claimed to be host to a WD with a mass close to the Chandrasekhar limit[36]. Furthermore, a prominent carbon-rich equatorial dusty disk such as that in V445 Pup (refs. 34,35) could explain (see 'V445 Puppis' section in Methods) the luminous infrared counterpart of SN 2020eyj (Extended Data Fig. 2), which we attribute to an infrared echo from radiatively

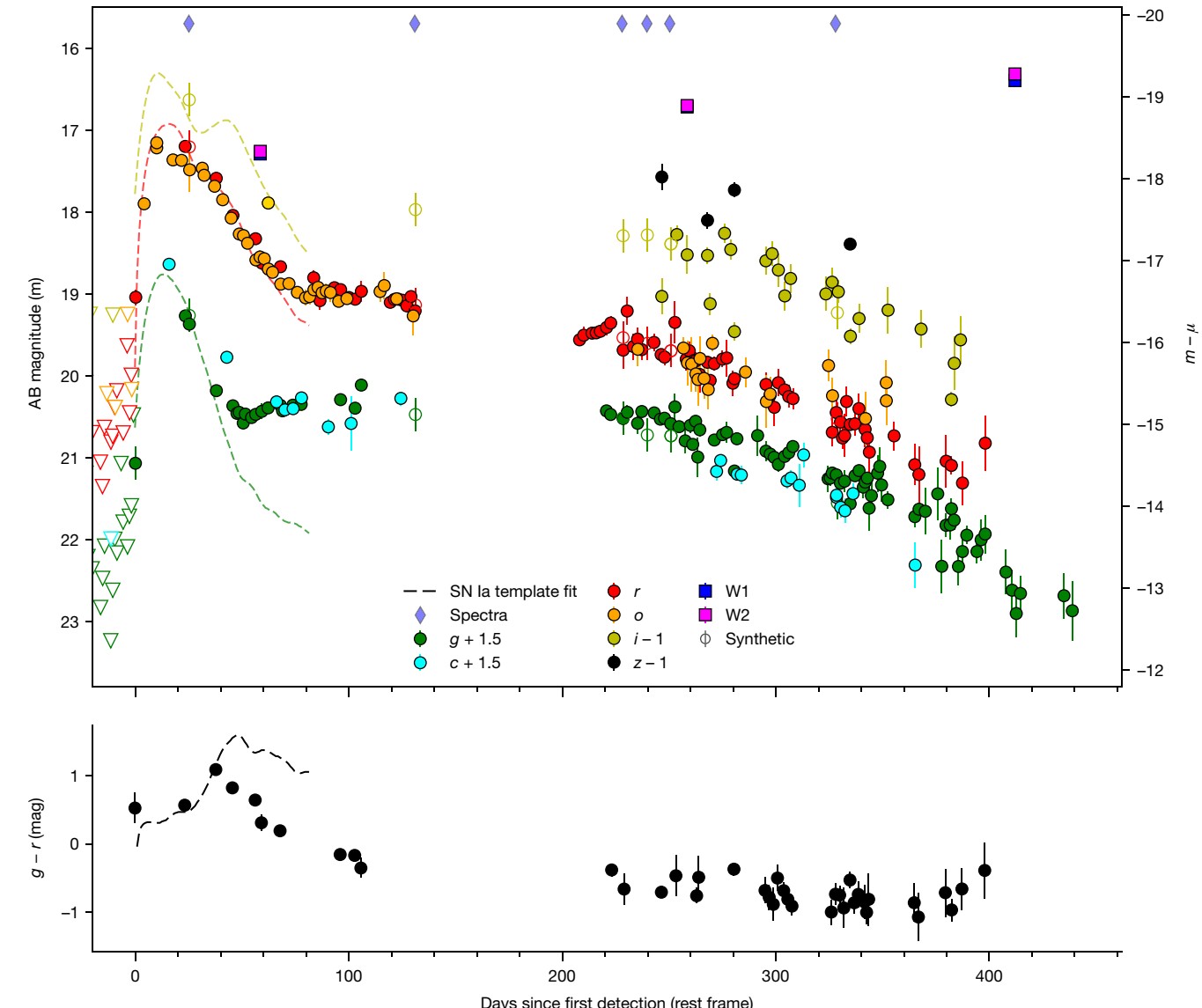

**Fig. 3 | The multiband light curve of SN 2020eyj can be divided into a diffusion-peak phase and a long-lived interaction-powered tail phase.** The light curves of SN 2020eyj are shown with overplotted SN Ia template fits to the initial peak (see 'Light-curve analysis' section in Methods). The most recent mid-infrared epoch (W1 and W2) is outside the date range plotted here and is shown in Extended Data Fig. 2. Open circles indicate synthetic photometry derived from the spectra. Phase is in rest-frame days since first detection. Apparent magnitudes on the left $y$ axis, absolute magnitudes on the right $y$ axis,

in which $\mu$ is the distance modulus. Non-detections with $5\sigma$ upper limits are indicated by triangles. The photometry has been binned into one-night bins and has been corrected for MW reddening. The diamond markers at the top indicate the epochs of spectroscopy. The bottom panel shows the $g - r$ colour for the nights in which both $g$ and $r$ photometry was obtained, overplotted with the colour evolution of a typical SN Ia. The error bars represent $1\sigma$ uncertainties.

heated pre-existing dust with a dust mass of order $10^{-2} M_\odot$ (see 'Dust properties' section in Methods). The initial models invoking recurrent novae for the origin of PTF11kx (ref. 15) were challenged by the CSM masses involved[37], which were too large by orders of magnitude for symbiotic nova mass build-up models[38]. Similarly, the mass resulting from a V445 Pup-like nova outburst ($\lesssim 10^{-3} M_\odot$; see 'V445 Puppis' section in Methods) is insufficient to explain the CSM mass observed in SN 2020eyj. However, a recent study of the radio evolution of V445 Pup suggests that the equatorial disk could have pre-dated (and survived) the nova outburst[39], which would allow for mass build-up in the disk between nova eruptions. This scenario would require the SN to occur soon after the nova outburst and before the resumption of mass transfer between the donor and WD reforms the disk at small radii. We note that a nova similar to the year 2000 event of V445 Pup would not have

been detectable at the distance of SN 2020eyj (see 'Precursor search' section in Methods).

SN 2020eyj represents the first observational example of the previously speculated class of SNe Ia–He CSM (ref. 40). The presence of a dense CSM, supported by a radio detection, offers strong evidence for the SD scenario for SN 2020eyj, in particular for the He star + WD formation channel. It is estimated that about 10% of all SD SNe Ia arise from this channel[7], which is probably the dominant source of SNe Ia with short delay times[41]. Understanding the timescale of SN Ia activity is important for the chemical evolution of galaxies. The confirmed presence of a He-rich CSM in a SN Ia system also affects SN Ia explosion modelling, as He plays a vital role in double detonation models in which the WD explosion is triggered by the ignition of a massive ($\lesssim 0.2 M_\odot$) He shell on its surface[30]. Constraining the rate of SNe Ia similar to

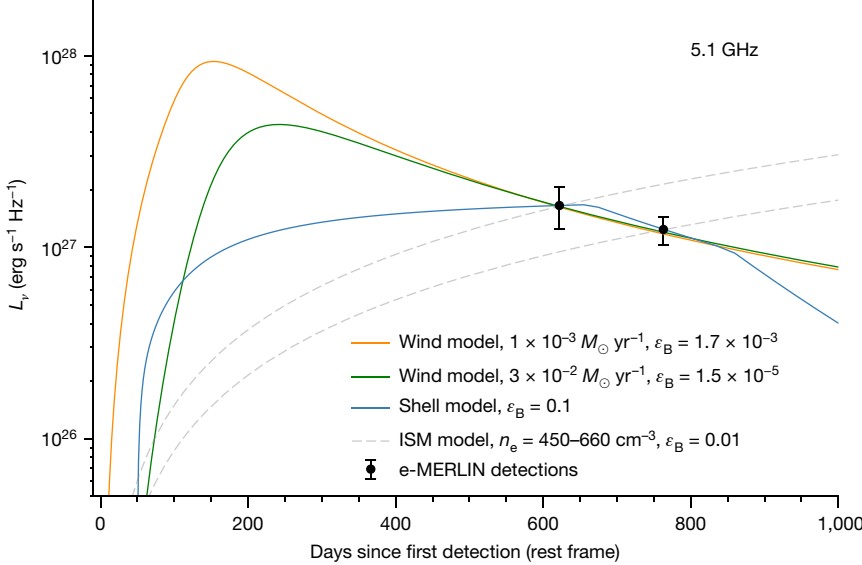

**Fig. 4 | The radio detections of SN 2020eyj at 5.1 GHz can be reconciled with CSM interaction.** For the wind model, in which the CSM follows a density profile of $\rho \propto r^{-2}$, we assume a pre-SN wind velocity of 1,000 km s$^{-1}$ and adopt a mass-transfer rate as inferred from fitting the bolometric light curve of SN 2020eyj. Depending on the level of line-of-sight extinction affecting the bolometric light curve (see 'Bolometric light curve' section in Methods), the wind model fits the observations (in black, with 1$\sigma$ uncertainties) well for the microphysics parameter $\epsilon_B = 1.7 \times 10^{-3}$ ($1.5 \times 10^{-5}$) and a CSM mass of $M_{csm} = 0.3\,M_\odot$ ($1\,M_\odot$) within $10^{17}$ cm (see 'Optically thick wind' section in Methods) when $E(B - V) = 0$ mag (0.5 mag). For the shell model, in which the CSM is concentrated in a constant-density CSM shell, we assume $\epsilon_B = 0.1$ and obtain a best estimate for the CSM mass of $M_{csm} = 0.36\,M_\odot$ and a CSM interaction end time of $t_{end} = 665$ days (a width of $8.6 \times 10^{16}$ cm; see 'CSM shells' section in Methods). In both the wind and shell model fits, $\epsilon_e = 0.1$ is assumed. We also show radio light curves from a model involving a DD SN Ia interacting with the ISM (see 'ISM' section in Methods). To fit the individual radio detections, this model requires unusually high ISM densities and neither fit reproduces the observed decline in flux, ruling out the DD scenario.

SN 2020eyj would require systematic spectroscopic follow-up of SNe Ia with long-lived light curves, as monitoring at present often stops after a seemingly normal SN Ia has been classified. Observational properties that SN 2020eyj share with its H-analogue PTF11kx, such as a fast rise and a 91T-like peak spectrum, can potentially guide such follow-up efforts and allow for the discovery and study of more SN 2020eyj-like SNe Ia, including at radio wavelengths.

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

# Methods

## Observations

**Discovery.** SN 2020eyj was discovered by the Asteroid Terrestrial-impact Last Alert System (ATLAS)[42,43] on 23 March 2020 UT (ref. 44) and subsequently detected as part of the Zwicky Transient Facility (ZTF) survey[45,46], at $\alpha$ = 11 h 11 min 47.19 s, $\delta$ = 29° 23′ 06.5″ (J2000). Pre-discovery detections were recovered in the ZTF data on 7 March 2020 UT (MJD = 58,915.12) in both $g$ and $r$ filters. For reference, we list some key characteristics of SN 2020eyj in Extended Data Table 1.

**Optical photometry.** Follow-up photometry was obtained as part of public and partnership ZTF survey observations[47] with the ZTF camera[48] on the Palomar 48-in. telescope (P48) telescope in the $g$ and $r$ bands, and later phases were also covered in the $i$ band. The P48 data were reduced and host subtracted using the ZTF reduction and image-subtraction pipeline[49], which makes use of the ZOGY algorithm[50] for reference-image subtraction. Following the rationale illustrated in ref. 51, we apply the difference image zero point magnitude to convert fluxes from units in detector data number (DN) to μJy and translate fluxes to AB magnitudes. We apply a detection threshold of signal-to-noise ratio (S/N) $\gtrsim$ 3 and for non-detections we compute $5\sigma$ upper limits. The Supplementary Information lists the ZTF magnitudes and upper limits.

Further photometric epochs were obtained with the Liverpool Telescope[52], the Spectral Energy Distribution Machine (SEDM) on the Palomar 60-in. telescope (P60), the Las Cumbres Observatory telescopes (programme ID NOAO2020B-012) and the Alhambra Faint Object Spectrograph and Camera (ALFOSC) on the Nordic Optical Telescope (NOT), with data reduced and host subtracted using the pipelines described in refs. 53,54 or standard methods. In this work, we also make use of the forced photometry service from the ATLAS survey[42,55], which contained valuable photometry in the $o$ and $c$ bands. One $i$-band epoch was obtained from the Pan-STARRS1 data archive[56].

ZTF and ATLAS also obtained observations of the location of SN 2020eyj in the nights immediately preceding the first detection, with limiting magnitudes in the ZTF $g$ band on 5 and 6 March 2020 UT of 20.8 and 19.7, respectively, and the (binned) observations in the $o$ band from ATLAS on 5 March 2020 UT correspond to a limiting magnitude of 20.2. Phases in this study are relative to the first ZTF detection (MJD = 58,915.212, 7 March 2020 UT) in rest-frame days, unless stated otherwise. Given the excellent constraints on the nights before, this epoch is also close to the time of first light.

All magnitudes are reported in the AB system. The extinction in the Milky Way (MW) was obtained from ref. 57 as $E(B - V)$ = 0.024 mag. MW reddening corrections are applied using the extinction law with $R_V$ = 3.1 (ref. 58), whereas SN reddening corrections are applied using $R_V$ = 2. The photometric magnitudes of SN 2020eyj are listed in the Supplementary Information. The ATLAS and P48 light curves are shown in Fig. 3, binned into one-night bins to enhance the S/N.

**Optical spectroscopy.** The first optical spectrum of SN 2020eyj was obtained with the SEDM (ref. 59) mounted on the P60 (ref. 60), 25 days after first detection. All SEDM spectra are automatically reduced and calibrated with pysedm (ref. 61) and the quality of the SEDM spectrum of SN 2020eyj was further improved using hypergal (ref. 62). Follow-up spectroscopy was obtained from 131 days onward with the Low-Resolution Imaging Spectrometer (LRIS)[63] on the Keck I telescope and the ALFOSC on the NOT (ref. 64). A host spectrum was obtained at 678 days, after the SN had fully faded from view. The spectra were reduced in a standard manner, using LPipe (ref. 65) and PypeIt[66,67] for Keck/LRIS and NOT/ALFOSC, respectively.

A log of the obtained spectra is provided in Extended Data Table 2 and the epochs of spectroscopy are indicated by the diamond markers on top of the light curves in Fig. 3. The spectra were absolute-flux-calibrated against the $r$-band magnitudes using the Gaussian process interpolated magnitudes and then corrected for MW extinction. All spectral data and corresponding information will be made available through the WISeREP public database[68]. We present the peak SEDM spectrum in Fig. 1 and the later sequence of spectra in Fig. 2.

The initial spectrum obtained with SEDM is characterized by broad absorption features (see 'SN Ia classification' section). The later spectra are shaped by broad Fe II lines, in particular the quasi-continuum blueward of 5,700 Å (refs. 69–73) and a prominent Ca II NIR triplet. Superimposed on the continuum are narrow He I emission lines, as well as Hα. We measure full width at half maximum (FWHM) velocities of the He I emission lines and Hα in the spectra obtained with Keck at 131 and 251 days by fitting a Lorentzian profile to the complete lines, as well as to just the blue wings. The red wings in the He and Hα lines are markedly attenuated (see Extended Data Fig. 3 and 'Dust properties' section), so the intrinsic FWHM velocities are better represented by (double) the blue-wing FWHM. We report these FWHM velocities in Extended Data Table 2. The FWHM velocities of the He I emission lines range from 1,100 to 2,700 km s$^{-1}$ (corrected for the red wings), with no sign of a narrow (<1,000 km$^{-1}$) component detected in some SNe Ibn and interpreted as coming from the unshocked CSM[71,74]. There is also no sign of material stripped from the donor star by the SN ejecta[75,76], which is predicted to show up as narrow emission (<1,000 km$^{-1}$ (ref. 77)).

The asymmetric line profile we associate with the SN also applies to the Hα emission line, suggesting the presence of H in the CSM. In the spectrum obtained at 131 days, Hα has an equivalent width of 14 Å, not corrected for contribution by the host. By comparison, the He I emission lines at 5,876 Å, 6,678 Å and 7,065 Å in the same spectrum have equivalent widths of 47 Å, 43 Å and 61 Å, respectively. As H is easier to ionize than He, the more prominent He lines means that the CSM must predominantly consist of He. By epoch 329 days, the Hα luminosity has dropped to the luminosity of the Hα narrow emission line in the host spectrum obtained at 678 days (see 'Host galaxy' section).

**Infrared photometry.** Following a report[78] of a mid-infrared detection of SN 2020eyj in the 2021 data release of the NEOWISE Reactivation (NEOWISE-R)[79] survey, we queried the IPAC Infrared Science Archive for any NEOWISE-R detections at the position of SN 2020eyj. After filtering poor-quality data and binning individual exposures following the method described in ref. 80, the SN was recovered in both W1 and W2 filters (3.4 and 4.6 μm, respectively) in all four 2020 and 2021 epochs, with the earliest detection at 59 days after first detection (Fig. 3 and Extended Data Table 3). The host is not detected in (stacked) WISE data before the SN explosion (Extended Data Fig. 2, top panels), so we assume that the contribution from the host is negligible and all observed flux is because of the SN.

**Radio.** We observed SN 2020eyj with the electronic Multi-Element Radio Linked Interferometer Network (e-MERLIN) in two epochs. The first epoch, with a duration on target and phase calibrator of about 16 h, was conducted on 19 November 2021 (centred on MJD 59,538.29), 605 days after first detection and included six e-MERLIN telescopes (Mk2, Kn, De, Cm, Da and Pi). The second epoch was conducted during six consecutive days between 6 and 12 April 2022 (mean MJD 59,678.59, 741 days after first detection). Between fix and six telescopes (including the Lovell) participated, with some antennae missing part of the runs owing to technical problems. Owing to the much smaller field of view of the Lovell telescope, the pointing centre of the second epoch was shifted by 1 arcmin to include an in-beam calibrator in the primary beam of this telescope. 3C 286 and OQ 208 were used as amplitude and band-pass calibrators, respectively. The phase calibrator, J1106+2812, was correlated at position $\alpha_{\rm J2000.0}$ = 11 h 06 min 07.2617 s and $\delta_{\rm J2000.0}$ = 28° 12′ 47.065″, at a separation of 1.7° from the target, and was detected with a flux density of 150 mJy. We centred our observations at a frequency of 5.1 GHz, using a bandwidth of 512 MHz. The data were correlated with the e-MERLIN correlator at Jodrell Bank Observatory,

using four spectral windows, each of 512 channels, with 1-s integrations and four polarizations.

We calibrated and processed the data using the e-MERLIN CASA pipeline[81] version v1.1.19 running on CASA version 5.6.2. We used the 10-mJy in-beam source to self-calibrate the residual phases and amplitudes of the target source. Cleaning was done with the software package wsclean[82]. Final images of the target were produced with a synthesized beam of 80 mas × 35 mas at a position angle of 28° and 94 mas × 71 mas at a position angle of −71°, in the first and second epochs, respectively. The $1\sigma$ root mean squares of the images is 17 and 8 μJy beam$^{-1}$, respectively. The target is detected in both epochs as an unresolved source as characterized with task IMFIT. We estimate the uncertainty of the peak flux density to be a quadratic sum of the image root mean square and a conservative 10% amplitude scale calibration error. The final flux density of the source is $80 \pm 20$ and $60 \pm 10$ μJy beam$^{-1}$ in the first and second epochs, respectively. The radio source is located at an average position of $\alpha_{J2000.0} = 11$ h 11 min 47.1763 s and $\delta_{J2000.0} = 29°$ 23′ 06.45″, with an estimated uncertainty of 10 mas.

The average position of the e-MERLIN detections relative to the optical positions of SN 2020eyj is shown in Extended Data Fig. 4. The radio detection is consistent with the position of the SN in the ALFOSC epoch at 382 days ($r$ band) and the position reported in GaiaAlerts of the detection of SN 2020eyj in the $G$ band at 42 days.

**X-ray.** On 27 April 2022, 758 days after first detection, we observed SN 2020eyj for 3.8 ks with the X-ray telescope XRT between 0.3 and 10 keV aboard the Neil Gehrels Swift Observatory[83,84]. We analysed the data with the online tools of the UK Swift team (https://www.swift.ac.uk/user_objects/) that use the methods described in refs. 85,86 and the software package HEASoft version 6.26.1. SN 2020eyj evaded detection down to a count rate of 0.003 count s$^{-1}$ ($3\sigma$ limit). To convert the count-rate limit into a flux limit, we assumed a power-law spectrum with a photon index $\Gamma$ of 2 and a galactic neutral hydrogen column density of $1.9 \times 10^{20}$ cm$^{-2}$ (ref. 87). Here the photon index $\Gamma$ is defined as the power-law index of the photon flux density ($N(E) \propto E^{-\Gamma}$). Between 0.3 and 10 keV, the count-rate limit corresponds to an unabsorbed flux of $1.1 \times 10^{-13}$ erg cm$^{-2}$ s$^{-1}$ and a luminosity $<2.4 \times 10^{41}$ erg s$^{-1}$. It is possible that a deeper observation would have yielded a detection, as the type Ia–CSM SN 2012ca was detected in X-rays at a similar epoch, with a luminosity on the order $10^{40}$ erg s$^{-1}$ (ref. 88).

## SN Ia classification

During the peak phase of SN 2020eyj, an optical spectrum was obtained with the low-resolution ($R \approx 100$) SEDM on the P60, 25 days after first detection. This high S/N spectrum was characterized by broad absorption features (Fig. 1), based on which SN 2020eyj was classified as a SN Ia at redshift $z = 0.03$ (ref. 8). Using SNIascore, a deep-learning-based classifier of SNe Ia based on low-resolution spectra[89], it was noted that the SN could be a type Ibc SN erroneously classified as SN Ia because of the degeneracy between peak spectra of SNe Ibc with those of SNe Ia at post-peak phases, but their classifier favoured a SN Ia classification anyway. In general, based on the comparison study in ref. 13, type Ibc SNe erroneously classified as type Ia(–CSM/91T) are much less common than the inverse. Here we scrutinize the SEDM spectrum using comparisons with SNe from the literature, based on spectral matching with the SuperNova IDentification (SNID)[90] and Superfit[91] classification tools, for which the SNID template library has been supplemented with the type Ibc templates from ref. 92. We adopt a $g$-band peak epoch of MJD = $58,929 \pm 2$, based on the light-curve fitting described in the 'Light-curve fit' section, with the uncertainty driven by the poor sampling of our photometry around peak.

The top 10 SNID (rlap > 10) and Superfit matches are all of type Ia (Fig. 1) and include normal SNe Ia such as SN 2004eo (ref. 93) and 91T-like SNe such as SN 2001V (ref. 94). The best matching SN of type Ibc (rlap = 8) is the type Ic SN 1994I (refs. 95–98). The phases corresponding

to the matched SNe Ia are all post-peak, ranging from 12 days to about 50 days post-maximum, whereas the matching SN Ibc spectra are all within a few days from peak. The phase of the SEDM spectrum of SN 2020eyj is 12 days post-maximum, which corroborates the SN Ia classification.

In terms of spectral features, the SEDM spectrum shows broad absorption lines that, based on the spectral comparisons, can be unambiguously identified as Si II, Fe II and Ca II (Fig. 1). Compared with normal SNe Ia as exemplified by SN 2004eo, the Si II features in SN 2020eyj are shallow. Diluted Si II absorption is common for 91T-like SNe Ia, as in the spectrum of SN 2001V. Type Ia–CSM are known to show 91T-like spectra around peak[13]. As a SN strongly interacting with a CSM, the presence of diluted Si II in the SEDM spectrum of SN 2020eyj is consistent with a type Ia(–CSM) classification. In terms of expansion velocity, the velocity of the Si II$\lambda$6355 absorption feature in the SEDM spectrum is $8,900 \pm 600$ km s$^{-1}$. This velocity is on the slow side for the SN Ia sample described in ref. 99 but consistent with the SN Ia sample in ref. 100 and comparable with, for example, SN 2004eo (Fig. 1).

Another notable feature in the SEDM spectrum is the complete lack of O I 7,774 Å absorption (Fig. 1), even though O I absorption in SNe Ia is common, in particular 91T-like SNe Ia can have shallow or non-existent O I (ref. 101). This is clearly visible in the matched spectrum of SN 2001V. By contrast, SNe Ibc that lack O I absorption are extremely uncommon, especially at about 12 days post-peak[102,103], as exemplified by type Ic SN 1994I in Extended Data Fig. 5. In this figure, we have also included the type Ibn SN 2006jc at a phase similar to that of the SEDM spectrum, to highlight that SN 2020eyj does not show any sign of He I emission lines or the quasi-continuum expected for a type Ibn at this phase.

An absence of oxygen lines is typical for type Ia–CSM spectra, both as an absorption feature around peak and as emission in later epochs[11,104], as seen in the early and late spectra of PTF11kx in Figs. 1 and 2, respectively. Similarly, the late spectra of SN 2020eyj lack any sign of O I$\lambda$7774 emission (Fig. 2). Other features in the late-time spectra of SN 2020eyj that are typical for type Ia–CSM include prominent broad Ca II emission and a high H$\alpha$/H$\beta$ Balmer ratio, which indicates that the emission lines are probably produced through collisional excitation rather than recombination[11]. The high S/N spectrum at 251 days shows both H$\alpha$ and H$\beta$ emission, but after correcting for contribution by the host, only H$\alpha$ shows some residual flux related to the transient. We note that, at this late phase, SN 2020eyj is about four magnitudes brighter than expected from a normal SN Ia, such as SN 2004eo (ref. 93), and the spectrum is dominated by CSM-interaction features.

In conclusion, based on its spectral features, we classify SN 2020eyj as a type Ia(–CSM) SN. Furthermore, as we discuss in the 'Light-curve fits' section, the light curves of SN 2020eyj show strong similarities to those of PTF11kx, the SN that cemented SNe Ia–CSM as a subclass.

## Light-curve analysis

**Light-curve fits.** The light curve of SN 2020eyj (Fig. 3) can be divided into two phases, similar to its spectral evolution. In the first phase, lasting around 50 days, the light curve follows a fairly typical bell-like shape, peaking at $m \approx 17.2$ in both the $r$ band and the ATLAS bands, which—at a luminosity distance of 131.4 Mpc (see 'Host galaxy' section)—corresponds to $M \approx -18.4$, not accounting for host extinction. During the first phase, the light curve has a red $g - r$ colour, consistent with the classification spectrum. The second phase, the tail phase from 50 days onward, is characterized by a slowly evolving light curve with spectra that are dominated by CSM interaction. Although the $r$-band light curve continues to fade, albeit at a slower rate of about 0.6 mag per 100 days between days 50 and 251, the $g$-band light curve plateaus. This results in a $g - r$ colour change to blue (see bottom panel of Fig. 3), which—based on the spectra—is driven by the pseudo-continuum blueward of 5,700 Å. This Fe II feature, typical for CSM-interaction-powered spectra, is well traced by the ZTF $g$ band (4,100–5,500 Å). From 251 days onward, the light curve fades in all bands at a rate of about 1 mag per 100 days.

The transition between the two phases is well captured by the photometry at 50 days, when the decline in the $g$ band is abruptly halted and changes to a plateau lasting around 200 days. This divergence of the $g$-band light curve from a smooth decline is probably the epoch in which CSM interaction starts contributing (substantially) to the light curve and in which the spectra start to look like those of SNe Ibn. But even though the late spectra may be similar to SNe Ibn, the light curve is unlike those of documented SNe Ibn. SNe Ibn are characterized by uniform, rapidly evolving blue light curves[105], peaking at $M_r \approx -19.5$. There is a handful of long-lived, slowly evolving SNe Ibn reported in the literature, but they are either much brighter than SN 2020eyj (refs. 106,107) or have a much longer risetime[108]. None of the literature SNe Ibn show a long-duration (>300 days), slowly evolving light-curve tail such as that observed in SN 2020eyj. It is worth noting that there have been suggestions in the literature that some SNe Ibn may come from thermonuclear explosions, hidden by a dense CSM[109]. The discovery of SN 2020eyj seemingly supports that notion.

The post-peak decline rates and peak magnitudes of SNe Ia are strongly correlated (the Phillips relation[110]), with brighter (fainter) SNe Ia evolving slower (faster). We fit the first phase of the multiband light curves with SNooPy[111], to determine whether the width (stretch) of SN 2020eyj is consistent with the expected peak luminosity. The light curve of SN 2020eyj up to 50 days is well described by a SN Ia light curve with an adopted stretch of $s_{BV} = 1.2 \pm 0.1$ and an extinction of $E(B - V) = 0.5 \pm 0.1$ mag (adopting a total-to-selective extinction ratio $R_V = 2.0$), resulting in a peak magnitude approximately 0.06 mag fainter than expected from the Phillips relation. The required line-of-sight extinction is considerable but is consistent with the host extinction of $E(B - V) = 0.54^{+0.14}_{-0.12}$ mag derived from host galaxy Balmer lines (see 'Host galaxy' section). We apply the same fitting method to the light curve of PTF11kx, consisting of published and previously unpublished photometry. For PTF11kx, we adopt the same stretch factor of 1.2 and obtain an extinction of $E(B - V) = 0.27 \pm 0.02$ mag, consistent with the extinction $A_V \approx 0.5$ mag derived in ref. 15. After correcting for the fitted host extinction, the resulting absolute-magnitude light curves of SN 2020eyj and PTF11kx are practically identical in the $g$ and $r$ bands for the first approximately 45 days, even though the fits are independent (Extended Data Fig. 1). The $r$-band light curves peak at $M_r \approx -19.3$ for both SNe, consistent with both SNe Ia and SNe Ia–CSM, although both SNe are on the fainter end of the sample of SNe Ia–CSM described in ref. 11. From the light-curve fits, we obtain for SN 2020eyj risetimes in the $g$ and $r$ bands of $14 \pm 2$ and $16 \pm 2$ days since first detection, respectively. This is fast for a SN Ia (ref. 33) but similar to PTF11kx (Extended Data Fig. 1).

An important caveat about the light-curve fit is that the intrinsic decline rate of SN 2020eyj could seem slower because of the contribution by CSM interaction. On the basis of the colour evolution of the light curve, we know from day 50 onward that the CSM contribution is notable, but it is reasonable to assume that some CSM interaction already contributes to the light curve at earlier epochs. This means that the stretch parameter we measure should be regarded as an upper limit and, as a result, so is the peak luminosity of the fit. SN 2020eyj, but also PTF11kx, are not typical SNe Ia, so the colours and peak magnitude could (to some extent) also be a property intrinsic to the class.

**Bolometric light curve.** The light curve of SN 2020eyj around peak has limited photometric coverage, in both wavelength and cadence, which hinders the construction of a precise, full bolometric light curve. Instead, we compute the bolometric light curve based on the SN Ia light-curve template fit obtained in the 'Light-curve fits' section, for epochs when the photometry (notably the $g$ band) still matches well with the fitted light curve (up to 38 days after first detection; Extended Data Fig. 1). From the fitted optical light curves, we flux calibrate, correct for host extinction and integrate the spectral time series from ref. 112 from the ultraviolet (UV) to the NIR (1,000–25,000 Å). For the

tail phase, we integrate the Keck spectra at 131 and 251 days from 3,000 to 10,000 Å and apply a linear extrapolation in the UV to zero flux at 2,000 Å. There is little spectroscopic (Fig. 2) and colour (Fig. 3) evolution between the Keck spectrum at 251 days and the NOT spectrum at 328 days, so we extend the pseudo-bolometric light curve to the final photometric epoch at 383 days obtained with ALFOSC on the NOT assuming a constant bolometric correction applied to the $g$-band magnitude. Extended Data Fig. 6 shows the bolometric luminosity inferred from the template fit, the Keck spectra and the final photometric epoch. The template fit to the initial peak included considerable line-of-sight extinction of $E(B - V) = 0.5$ mag (see 'Light-curve fits' section). To account for the possibility that SN 2020eyj may be intrinsically faint rather than a normal SN Ia substantially dust-extincted, we also include a bolometric light curve for $E(B - V) = 0$. Depending on dust extinction, the total integrated energy radiated across the bolometric light curve amounts to $0.6–1.2 \times 10^{50}$ erg.

### Host galaxy
The host of SN 2020eyj is a faint and compact galaxy with designation SDSS J111147.15+292305.9 (Extended Data Fig. 4). We retrieved science-ready co-added images from the Galaxy Evolution Explorer (GALEX) general release 6/7 (ref. 113), the Sloan Digital Sky Survey data release 9 (SDSS DR 9; ref. 114) and the Panoramic Survey Telescope and Rapid Response System (Pan-STARRS, PS1) DR1 (ref. 56) and measured the brightness of the host using LAMBDAR (Lambda Adaptive Multi-Band Deblending Algorithm in R; ref. 115) and the methods described in ref. 116. We augment this dataset with an optical $r$-band image obtained with ALFOSC on the NOT on 2 May 2022 and UV observations from Swift/UVOT from 27 April 2022. The photometry on the UVOT images was done with uvotsource in HEASoft and an aperture encircling the entire galaxy (aperture radius 8″). Extended Data Table 4 lists all measurements. We fit the host galaxy spectral energy distribution (SED) with the software package Prospector version 0.3 (ref. 117) to determine the host galaxy properties. We assumed a Chabrier initial mass function[118] and approximated the star formation history (SFH) by a linearly increasing SFH at early times followed by an exponential decline at late times (functional form $t \times \exp(-t/\tau)$, in which $t$ is the age of the SFH episode and $\tau$ is the $e$-folding timescale). The model was attenuated with the model in ref. 119. The priors were set identical to those in ref. 116. The fit resulted in a low host-galaxy mass of $\log(M/M_\odot) = 7.79^{+0.15}_{-0.34}$.

We obtained a host galaxy spectrum with LRIS/Keck after SN 2020eyj had faded from view, at 678 days. We identify unresolved ($\lesssim 150$ km s$^{-1}$) host-galaxy lines in the spectrum, such as numerous Balmer lines in emission and absorption, [N II] $\lambda\lambda 6548,6583$, [O II] $\lambda\lambda 3726,3729$, [O III] $\lambda\lambda 4959,5007$ and [S II] $\lambda\lambda 6716,6731$, based on which we measure a redshift of $z = 0.0297 \pm 0.0001$. Adopting a flat cosmology with $H_0 = 70$ km s$^{-1}$ Mpc$^{-1}$ and $\Omega_M = 0.3$, this redshift corresponds to a luminosity distance to SN 2020eyj of 131.4 Mpc, which we use throughout this paper. Correcting for MW extinction, the adopted distance results in a host galaxy absolute magnitude of $M_r = -15.8$.

On the basis of the Balmer decrement measured in the host spectrum, we estimate a host extinction with $E(B - V) = 0.54^{+0.14}_{-0.12}$ mag, in agreement with the extinction obtained by fitting the light curves of SN 2020eyj with a SN Ia template (see 'Light-curve fits' section). The line ratios of $\log_{10}([O III] \lambda 5007/H\beta) = 0.39$ and $\log_{10}([N II] \lambda 6583/H\alpha) = -1.26$ put the host galaxy well into the regime of star-forming galaxies on the Baldwin–Phillips–Terlevich diagram[120]. Adopting the parameterization of the empirical oxygen calibration O3N2 by ref. 121, we obtain an oxygen abundance of $12 + \log(O/H) = 8.14 \pm 0.03$. Such a low oxygen abundance is expected for a low-mass galaxy[122].

The host properties of 16 SNe Ia–CSM were reported in refs. 11,123. The authors concluded that all objects in their samples exploded in star-forming late-type galaxies (spiral and dwarf galaxies) with absolute magnitudes between $M_r = -20.6$ and $-18.1$ mag. The hosts of three SNe in this sample evaded detection in archival SDSS images, implying an

absolute magnitude $M_r > -18$ mag. SN 2020eyj exploded in a markedly low-luminosity star-forming dwarf galaxy with an absolute $r$-band magnitude of only $M_r = -15.8$ mag (not corrected for host attenuation). However, the modelling of the host-galaxy SED and the Balmer decrement show non-negligible attenuation of $0 < E(B - V) < 0.55$ mag ($3\sigma$ confidence interval from host SED modelling) or $0.2 < E(B - V) < 1$ mag ($3\sigma$ confidence interval from the Balmer decrement), which would alleviate the apparent extremeness of the host galaxy.

## Dust properties

Infrared emission is commonly observed in interacting SNe and can be attributed to the condensation of dust in the SN ejecta or in the shocked CSM, or to pre-existing dust in the CSM that is heated radiatively by the SN emission or by the ejecta–CSM shock interaction[124–127]. The mid-infrared luminosity of SN 2020eyj is at a similar level as for the most infrared-luminous interacting SNe, such as type IIn– and Ia–CSM SNe, and at 4.5 μm is 6–10 magnitudes brighter than normal type Ia SNe and ≳4 magnitudes brighter than the type Ibn SN 2006jc (Extended Data Fig. 2, bottom panel).

Assuming optically thin dust, the flux $F_\nu$ can be written as[128]:

$$F_\nu = \frac{M_d \, B_\nu(T_d) \kappa_\nu(a)}{d^2}, \tag{1}$$

in which $M_d$ is the mass of the dust, $B_\nu$ the Planck blackbody function, $T_d$ the temperature of the dust, $\kappa_\nu(a)$ the dust absorption coefficient as a function of dust particle radius $a$ and $d$ is the distance to the observer. For simplicity, we assume a simple dust population of a single size composed entirely of amorphous carbon with a grain size of 0.1 μm with the corresponding absorption coefficient $\kappa$ as in refs. 129,130 and fit the WISE data to obtain an estimate of the dust temperature and mass. We note that the dust mass depends on assumed grain size, which we cannot constrain on the available data. Varying the grain size from 0.01 to 1.0 μm changes the derived dust mass by an order of magnitude[129]. Over the first three epochs, up to 412 days, we derive a constant dust temperature of around 800 K (Extended Data Table 3), consistent with a lack of colour evolution in the WISE photometry (Fig. 3). Only at the fourth WISE epoch (614 days) do we see a marked decrease in the dust temperature, to $608 \pm 23$ K. These dust temperatures are well below the expected evaporation temperature of dust (1,500 K for silicates and 1,900 K for graphite grains[127]). As well as the dust temperatures, we obtain dust mass estimates of $(1.8 \pm 0.3) \times 10^{-3} \, M_\odot$ to $(9.9 \pm 2.1) \times 10^{-3} \, M_\odot$ for the first and final WISE epochs, respectively (Extended Data Table 3). The dust mass estimated for the final epoch corresponds to a CSM mass of $1 \times (0.01/r_{dg}) \, M_\odot$, in which $r_{dg}$ is the dust-to-gas ratio. The total integrated energy emitted in the infrared is $9 \times 10^{49}$ erg (Extended Data Table 3), which is similar to the integrated energy emitted in the optical (see 'Bolometric light curve' section).

In the case of optically thin dust that we consider here, the blackbody radius can be interpreted as a lower limit to the radius at which the dust resides. In the case of SN 2020eyj, the blackbody radius is $(2.5 \pm 0.2) \times 10^{16}$ cm in the first epoch and increases thereafter to $(6.4 \pm 0.6) \times 10^{16}$ cm at 614 days (Extended Data Table 3). Assuming a SN ejecta velocity of $10^4$ km s$^{-1}$, by 59 days, the ejecta would only have travelled about 20% of the distance inferred from the blackbody fit at that epoch. Combined with the constant dust temperature, this suggests that the infrared emission of SN 2020eyj is dominated by pre-existing dust being radiatively heated by CSM interaction emission, as was the case in type Ia–CSM SN 2005gj (ref. 130). Furthermore, because the dust reached a temperature of 800 K as early as 59 days and showed no notable evolution afterwards, it is unlikely that any surrounding dust was evaporated owing to the SN, because such hot dust would have dominated the infrared flux. For a peak SN luminosity of roughly $10^{43}$ erg s$^{-1}$ (Extended Data Fig. 6), the dust evaporation radius is $R_{evap} = (0.34–2.6) \times 10^{17}$ cm, depending on dust grain size and composition[127]. The lack of dust at the sublimation temperature implies

that the immediate region surrounding the SN was devoid of dust, much like the CSM-free cavity inferred from the SN light curve.

The He I and Hα emission-line profiles show the red wing being attenuated with time (Extended Data Fig. 3). Such an evolution in the line profiles has been interpreted to result from condensation of dust in the ejecta or the shocked CSM, obscuring the red wing of the emission line[126,131,132]. Similar line profiles have been observed in many SNe Ia–CSM[11] and in the prototypical type Ibn SN 2006jc, in which the evolution of the line profiles was attributed to dust condensing in a cool dense shell produced by the interaction of the ejecta with CSM also producing a substantial infrared excess[125]. Notably, such line-profile evolution has also been observed in the He nova V445 Pup, which was attributed to dust obscuration within the shell[35]. In particular, for type Ia–CSM SN 2005gj, dust formation was inferred from line profiles[11], whereas the bulk of the infrared emission was also attributed to pre-existing dust[130]. Although the line asymmetry in the spectra of SN 2020eyj is consistent with dust formation, it must be noted that asymmetric line profiles can also arise from optical depth effects, for example, in SNe Ibn (ref. 20). A substantial contribution to the infrared flux by newly formed dust is also not consistent with the lack of colour evolution in the light curve of SN 2020eyj from day 100 onward. The dust formation in SN 2006jc was accompanied by a reddening of the optical light curve[125], which we do not observe in SN 2020eyj past 100 days. So, we attribute the bright infrared counterpart of SN 2020eyj predominantly to pre-existing dust, similar to the conclusion drawn for the infrared counterpart of the prototypical type Ia–CSM SN 2002ic (ref. 124).

## CSM origin

**Optically thick wind.** Using progenitor models for the He star donor SN Ia channel from ref. 6, Moriya et al.[26] investigated the CSM properties resulting from this channel, in which accretion from a non-degenerate He star allows the accompanying WD to reach the Chandrasekhar limit. The study by Moriya et al.[26] focused on the low-circumstellar-density regime, in which the CSM properties in the WD + He star systems still adhere to the stringent CSM constraints imposed by radio non-detections of SNe Ia (refs. 4,5,133,134). Here we explore if the models with sufficiently dense CSM, with a wind-like density profile ($\rho \propto r^{-2}$), can explain the interaction-powered light-curve tail of SN 2020eyj and the detections at radio wavelengths. To quantify the properties of the CSM, we fit the CSM-interaction-powered tail of the bolometric light curve using the analytical model from ref. 26 and use the resulting mass-transfer rates to fit the radio detections.

Extended Data Fig. 6 shows the bolometric light curve of SN 2020eyj, with the initial peak described by the SN Ia template fit (solid line), and for the tail phase the luminosities inferred from the Keck spectra at 131 and 251 days and the ALFOSC epoch at 383 days (see 'Light-curve fits' section). From the light curve described by the SN Ia component alone (solid and dotted lines), it is also clear that the late-time light curve of SN 2020eyj cannot be powered by $^{56}$Ni decay, as the flux integrated across the Keck spectrum at 131 days is already at least ten times larger than what the radioactive decay delivers. Also plotted are the CSM-interaction model fits to the light-curve tail, for both $E(B - V) = 0$ and 0.5 mag, as discussed in the 'Light-curve fits' section. Assuming a pre-SN wind velocity of 1,000 km s$^{-1}$, the CSM-powered tail of SN 2020eyj is consistent with mass-transfer rates between $10^{-3} \, M_\odot$ yr$^{-1}$ ($E(B - V) = 0$ mag) and $3 \times 10^{-2} \, M_\odot$ yr$^{-1}$ ($E(B - V) = 0.5$ mag), which is 1–2 orders of magnitude larger than that considered in the original study[26]. At these very high mass-transfer rates, the critical mass accretion rate by the WD is exceeded and the excess is ejected as an optically thick wind, resulting in an extended He envelope[26]. In the model, the forward shock reaches approximately $10^{17}$ cm in 800 days. If we assume a wind velocity of 1,000 km s$^{-1}$, the CSM mass within $10^{17}$ cm in the models range from $0.3 \, M_\odot$ to $1 \, M_\odot$, for $E(B - V) = 0$ and 0.5 mag, respectively.

Extended Data Fig. 6 shows the wind model radio light curves fitted to the radio detections at 5.1 GHz, adopting $\epsilon_e = 0.1$ and mass-transfer

rates of $10^{-3}$ $M_\odot$ yr$^{-1}$ and $3 \times 10^{-2}$ $M_\odot$ yr$^{-1}$, for $E(B - V) = 0$ and 0.5 mag, respectively. We consider both synchrotron emission with synchrotron self-absorption and free–free absorption but note that, at the late phase of the radio detection, free–free absorption has only a minor impact. The radio light curve with an adopted mass-transfer rate of $10^{-3}$ $M_\odot$ yr$^{-1}$ is consistent with the radio detections of SN 2020eyj at 5.1 GHz, with microphysics parameter $\epsilon_B = 1.7 \times 10^{-3}$. For the high-extinction scenario, with a mass-transfer rate of $3 \times 10^{-2}$ $M_\odot$ yr$^{-1}$, the model fits when $\epsilon_B = 1.5 \times 10^{-5}$. In either case, the late time evolution follows the observed power-law decline rate of the observed radio luminosity of $\beta = -1.6$, which is comparable with that for hydrogen-free SNe Ibc (ref. 135).

It is worth noting that the bolometric light curve only extends to 400 days, whereas the first detection of SN 2020eyj at 5 GHz took place at 605 days. Furthermore, it has been argued that the mass-transfer rates associated with the optically thick wind phase ($>10^{-7}$ $M_\odot$ yr$^{-1}$) do not lead to SNe Ia but rather to accretion-induced collapse of the WD (refs. 136,137), although alternative wind models have been suggested to overcome this problem[138].

**CSM shells.** The CSM surrounding the H-rich analogue of SN 2020eyj, PTF11kx, was argued to be concentrated in shells[15]. Other SNe Ia have shown evidence for CSM concentrated in thin shells, albeit at distances (about $10^{16}$ cm) that no interaction with the ejecta is expected[139–142]. Shells have also been invoked for the configuration of the CSM in core-collapse H-rich type IIn SNe and typically attributed to ejection events by their massive progenitors. One noteworthy example is the well-studied SN 2014C, which transitioned from a stripped-envelope SN to a type IIn SN owing to interaction with a distant shell, and was detected in the radio[143,144]. Models for the radio emission of SNe Ia colliding with a constant-density shell of CSM have been previously presented in the literature, along with approximate functional forms to describe the evolution of the optically thick synchrotron light curve[145]. Because those models assume hydrogen-rich material, for our calculations, we modify $n_e = \rho/m_p$ to $n_e = \rho/(2m_p)$; otherwise, we use the default parameters, notably $\epsilon_B = 0.1$. We explore shell models with a range of CSM masses $M_{csm} = (0.01–1)$ $M_\odot$ and interaction end times from $t_{end} = 328$ days (the spectrum that does not show prominent He I lines) to $t_{end} = 763$ days (the second radio detection)—in this model, interaction must have ended before the second radio detection for the radio emission to have declined between the two observations. We assume a shell inner radius of $R_{in} = (30{,}000$ km s$^{-1}$)$(50$ days$) = 1.3 \times 10^{16}$ cm to close the system of equations in the model; then, the ranges of $M_{csm}$ and $t_{end}$ correspond to a range of shell widths $\Delta R/R_{in} = 3.4–7.5$. For each model, we calculate the representative model error as $\sigma_{mod} = \max(|L_{\nu,obs}(t_i) - L_{\nu,mod}(t_i)|/|\Delta L_{\nu,obs}(t_i)|)$, in which subscripts 'obs' and 'mod' refer to observed and modelled values, respectively, $L_\nu$ is spectral luminosity and $\Delta L_\nu$ is the error on the luminosity (flux error only; error in distance is not included). The best-fit model by this metric has $M_{csm} = 0.36$ $M_\odot$ and $t_{end} = 665$ days, which is a very similar mass to what is found for PTF11kx based on analysis of its optical spectra[146]. We find that models with $\sigma_{mod} \leq 3$ have $t_{end} \approx (500–763)$ days and $M_{csm} \approx (0.2–0.5)$ $M_\odot$, whereas those with $\sigma_{mod} \leq 1$ (that is, a better fit) have $t_{end} \gtrsim 580$ days and $M_{csm} \approx (0.3–0.4)$ $M_\odot$. The best-fit shell model is shown in Fig. 4.

**V445 Puppis.** The nova outburst of V445 Pup in the year 2000 lacked any Balmer emission in the spectra of its ejecta, but instead was characterized by He and carbon emission lines[34,147], making it the first and so far only known He nova system. On the basis of light-curve modelling, a mass ($\geq 1.35$ $M_\odot$) close to the Chandrasekhar limit was inferred for the WD in V445 Pup (ref. 36), consistent with the observed high ejecta velocities up to 8,450 km s$^{-1}$ (ref. 35). Combined with a high mass-transfer rate $>10^{-7}$ $M_\odot$ yr$^{-1}$, in which half of the accreted matter remains on the WD (ref. 36), V445 Pup is considered to be a prime candidate progenitor for the SD He + WD SN Ia progenitor channel.

On the basis of infrared spectra showing prominent carbon lines[34,148] and a rapid decline in the light curve of V445 Pup, it was shown that a carbon-rich thick dust shell must have formed in the nova ejecta[34,147]. High-resolution NIR images resolved the nova event into an expanding narrow bipolar shell with bulk velocities of about 6,700 km s$^{-1}$ and a perpendicular central dust disk that strongly attenuates the optical He I emission lines arising from the receding shell[35]. Seven years after the outburst, the bipolar shell of V445 Pup, as imaged in the NIR, extended to around $10^{17}$ cm and the central dust torus had an outer radius (perpendicular to the lobes) of $\geq 10^{16}$ cm (ref. 35). An outer dust shell in a V445 Pup-like system could survive dust sublimation from a SN Ia explosion, depending on peak luminosity and grain composition[127]. A recent study of the long-lived radio evolution of V445 Pup showed that the system was continuously synchrotron luminous for years after the outburst[39]. The synchrotron emission originated from the inner edge of the equatorial disk and was interpreted as interaction between a wind coming off the WD from nuclear burning and the surviving disk. The persistence of the disk through the nova outburst suggests that the disk is at least comparable in mass with the mass of the nova ejecta, which was estimated to be about $10^{-4}$ $M_\odot$ (ref. 36). In turn, the mass of the WD in V445 Pup, close to the Chandrasekhar limit, limits the ejecta mass in the system to not more than about $10^{-3}$ $M_\odot$ (Fig. 7 in ref. 36).

**ISM.** Radio emission can potentially arise from a SN Ia in the DD scenario as a result of interaction with the ISM. We have modelled the radio light curve from such a merger scenario in the same way as in refs. 31,134, that is, we assume that the supernova is the result of two merging WDs with masses 0.9 and 1.1 $M_\odot$, as described in ref. 149. The outermost ejecta has a density slope $\propto \rho^{-n}$ with $n = 13$ (see ref. 31 for a discussion on $n$). The microphysics parameters are the standard values $\epsilon_e = 0.1$ and $\epsilon_B = 0.01$. The modelled radio emission increases with time (Fig. 4) and, to agree with the observed 5.1-GHz fluxes at 605 and 741 days, the ISM electron density has to be 660 cm$^{-3}$ and 450 cm$^{-3}$, respectively, assuming fully ionized hydrogen and helium with He/H = 0.1. For $n = 13$ and fixed $\epsilon_e$, the electron density scales roughly as $\epsilon_B^{-0.74}$, so other ISM densities are possible accordingly. For a probable upper limit on $\epsilon_B$ of 0.1, the ISM density would be $n_e = 85$ cm$^{-3}$ to fit the flux at the second epoch, and for $\epsilon_B$ of 0.001, $n_e = 2{,}570$ cm$^{-3}$. The increase in radio flux with time is opposite to what is observed and is a property for all our ISM models with $n > 7.1$. Lower $n$ values are not expected[31] and the densities required in our ISM models are much higher than normal ISM densities. Moreover, for the $n = 13$, $\epsilon_B = 0.01$ model, in which $n_e = 450$ cm$^{-3}$, the modelled flux for the first epoch undershoots by $2\sigma$ (Fig. 4). In summary, our radio observations and their modelling argue strongly against an ISM scenario, which arises from a DD progenitor system. Furthermore, the observed strong helium lines are also at odds with an ISM scenario[150]. We therefore conclude that SN 2020eyj did not result from the thermonuclear runaway of a WD in a DD progenitor system, leaving the SD scenario as the only viable alternative.

**Precursor search.** The CSM surrounding SN 2020eyj could have originated from one or more novae, such as observed in V445 Pup. We investigate if a similar outburst at the location of SN 2020eyj can be found in ZTF data going back more than 2 years. The position of SN 2020eyj was observed 772 times (after quality cuts) in the $g$, $r$ and $i$ bands across 202 different nights in the final 2.29 years before the SN explosion. There are no notable pre-explosion detections in unbinned or binned light curves (1-day-long to 90-day-long bins) following the search method described in ref. 151. When combining observations in week-long bins, we reach a median limiting absolute magnitude of −14.28 in the $r$ band (−14.26 in the $g$ band). We can, hence, rule out precursors that are brighter than −14 magnitude 21% of the time in the $r$ band (16% of the time in the $g$ band). Precursors brighter than magnitude −15 can be ruled out 49% of the time in the $r$ band (39% for the $g$ band) in the final 2.29 years before the SN. The nova outburst of

V445 Pup peaked at $m_V = 8.6$ (ref. 152), which—at a distance of 8.2 kpc (ref. 35)—equates to an absolute magnitude of $M_V = -1$, far below the detection threshold of the ZTF.

## Data availability

The optical spectra of SN 2020eyj that support the findings of this study have been made available through the WISeREP archive (https://www.wiserep.org/object/14508). The ZTF photometry is listed in the Supplementary Information. Radio data from the electronic Multi-Element Radio Linked Interferometer Network (e-MERLIN) have been made available on the Zenodo repository with identifier https://doi.org/10.5281/zenodo.7665246. Data from the NEOWISE-R mission are available from the NASA/IPAC Infrared Science Archive with identifier https://doi.org/10.26131/IRSA124. Photometry from the Asteroid Terrestrial-impact Last Alert System (ATLAS) were obtained from a public source (https://fallingstar-data.com/forcedphot/).

## Code availability

On request, the corresponding author will provide the code used to produce the figures. The details of the models used in the 'Light-curve fits' and 'Optically thick wind' sections can be found in refs. 26,31,145 and references therein.

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

**Acknowledgements** We thank L. Dessart, M. Barlow and J. Nordin for helpful feedback and discussions and T. Szalai for providing us with their mid-infrared Spitzer light curves. E.C.K. acknowledges support from the G.R.E.A.T. research environment, financed by Vetenskapsrådet, the Swedish Research Council, project number 2016-06012; the research project grant 'Understanding the Dynamic Universe' financed by the Knut and Alice Wallenberg Foundation under Dnr KAW 2018.0067; and The Wenner-Gren Foundations. J.M. and M.P.-T. acknowledge financial support from the grant CEX2021-001131-S, funded by MCIN/AEI/ 10.13039/501100011033, and from the grant IAA4SKA (ref. R18-RT-3082) from the Economic Transformation, Industry, Knowledge and Universities Council of the Regional Government of Andalusia and the European Regional Development Fund from the European Union. M.P.-T also acknowledges financial support through grant PID2020-117404GB-C21, funded by MCIN/AEI/ 10.13039/501100011033. J.M. also acknowledges financial support from the grant PID2021-123930OB-C21 funded by MCIN/AEI/ 10.13039/501100011033, by "ERDF A way of making Europe" and by the "European Union". J.M. and M.P.-T. acknowledge the Spanish Prototype of an SRC (SPSRC) service and support financed by the Spanish Ministry of Science, Innovation and Universities, by the Regional Government of Andalusia, by the European Regional Development Funds and by the European Union NextGenerationEU/PRTR. T.J.M. is supported by the Grants-in-Aid for Scientific Research of the Japan Society for the Promotion of Science (JP20H00174, JP21K13966, JP21H04997). L.C. and C.H. acknowledge support from NSF grants AST-1751874, AST-1907790 and AST-2107070. S.M. acknowledges support from the Academy of Finland project 350458. The research of A.G.-Y. is supported by the EU through European Research Council (ERC) grant no. 725161, the ISF GW excellence centre, an IMOS space infrastructure grant and BSF/Transformative and GIF grants, as well as the André Deloro Institute for Advanced Research in Space and Optics, the Schwartz/Reisman Collaborative Science Program and the Norman E Alexander Family M Foundation ULTRASAT Data Center Fund, The Kimmel Center for Planetary Sciences, Minerva and Yeda-Sela; A.G.-Y. is the incumbent of the Arlyn Imberman Professorial Chair. N.L.S. is financed by the Deutsche Forschungsgemeinschaft (DFG, German Research Foundation) through the Walter Benjamin programme—461903330. K.M. is supported by H2020 ERC Starting Grant no. 758638 (SUPERSTARS). E.R. has received funding from the ERC under the European Union's Horizon 2020 research and innovation programme (grant agreement no 759194—USNAC). Based on observations obtained with the Samuel Oschin Telescope 48-in. and the 60-in. Telescope at the Palomar Observatory as part of the Zwicky Transient Facility (ZTF) project. The ZTF is supported by the National Science Foundation under grant no. AST-2034437 and a collaboration including Caltech, IPAC, the Weizmann Institute of Science, the Oskar Klein Centre at Stockholm University, the University of Maryland, Deutsches Elektronen-Synchrotron and Humboldt University, the TANGO Consortium of Taiwan, the University of Wisconsin at Milwaukee, Trinity College Dublin, Lawrence Livermore National Laboratories, IN2P3, France, the University of Warwick, the University of Bochum and Northwestern University. Operations are conducted by COO, IPAC and UW. This work was supported by the GROWTH Marshal project[153] financed by the National Science Foundation under grant no. 1545949. SED Machine is based on work supported by the National Science Foundation under grant no. 1106171. The ZTF forced-photometry service was financed under the Heising-Simons Foundation grant no. 12540303 (principal investigator: M. Graham). Based on observations made with the Nordic Optical Telescope, owned in collaboration by the University of Turku and Aarhus University, and operated jointly by Aarhus University, the University of Turku and the University of Oslo, representing Denmark, Finland and Norway, the University of Iceland and Stockholm University at the Observatorio del Roque de los Muchachos, La Palma, Spain, of the Instituto de Astrofisica de Canarias. The data presented here were obtained in part with ALFOSC, which is provided by the Instituto de Astrofisica de Andalucia (IAA) under a joint agreement with the University of Copenhagen and NOT. The Liverpool Telescope is operated on the island of La Palma by Liverpool John Moores University in the Spanish Observatorio del Roque de los Muchachos of the Instituto de Astrofisica de Canarias with financial support from the UK Science and Technology Facilities Council. e-MERLIN is a National Facility operated by the University of Manchester at Jodrell Bank Observatory on behalf of STFC. This work has made use of data from the Asteroid Terrestrial-impact Last Alert System (ATLAS) project. The ATLAS project is primarily funded to search for near-Earth asteroids through NASA grants NN12AR55G, 80NSSC18K0284 and 80NSSC18K1575; by-products of the NEO search include images and catalogues from the survey area. This work was partially financed by Kepler/K2 grant J1944/80NSSC19K0112 and HST GO-15889 and STFC grants ST/T000198/1 and ST/S006109/1. The ATLAS science products have been made possible through the contributions of the University of Hawaii Institute for Astronomy, Queen's University Belfast, the Space Telescope Science Institute, the South African Astronomical Observatory and the Millennium Institute of Astrophysics (MAS), Chile. The Pan-STARRS1 Surveys (PS1) and the PS1 public science archive have been made possible through contributions by the Institute for Astronomy, the University of Hawaii, the Pan-STARRS Project Office, the Max Planck Society and its participating institutes, the Max Planck Institute for Astronomy, Heidelberg and the Max Planck

Institute for Extraterrestrial Physics, Garching, Johns Hopkins University, Durham University, the University of Edinburgh, Queen's University Belfast, the Harvard–Smithsonian Center for Astrophysics, the Las Cumbres Observatory Global Telescope Network Incorporated, the National Central University of Taiwan, the Space Telescope Science Institute, the National Aeronautics and Space Administration under grant no. NNX08AR22G issued through the Planetary Science Division of the NASA Science Mission Directorate, the National Science Foundation grant no. AST-1238877, the University of Maryland, Eötvös Loránd University (ELTE), the Los Alamos National Laboratory and the Gordon and Betty Moore Foundation. We acknowledge ESA Gaia, DPAC and the Photometric Science Alerts Team (http://gsaweb.ast.cam.ac.uk/alerts).

**Author contributions** E.C.K. led the follow-up observations and is the primary author of the manuscript. J.J. conducted the spectral analysis and SN Ia light-curve modelling, and contributed to the source and infrared analysis. J.S. contributed substantially to the writing of the manuscript and the source analysis, and conducted follow-up observations with the NOT. J.M. and M.P.-T. led the radio observations and data analysis. T.J.M., L.C., C.H. and P.L. conducted the radio light-curve modelling. S.S. conducted the host-galaxy analysis. M.G. conducted follow-up observations with Keck. S.M. contributed to the writing and the infrared interpretation. S.Y. contributed to the data analysis. D.A.P. conducted follow-up observations with the Liverpool Telescope. N.L.S. conducted the precursor search. C.F., K.D. and Y.S. conducted follow-up observations. A.G.-Y. contributed to the writing and source analysis. J.L. and M.S. conducted the SEDM spectrum analysis. K.M., C.O., T.M.R. and S.D.R. contributed to the writing and source analysis. I.A., E.C.B., J.S.B., S.L.G., M.M.K., F.J.M., M.S.M., S.P., J.P., R.R. and D.S. are ZTF builders. All authors contributed to edits to the manuscript.

**Funding** Open access funding provided by Stockholm University.

**Competing interests** The authors declare no competing interests.

**Additional information**
**Correspondence and requests for materials** should be addressed to Erik C. Kool.

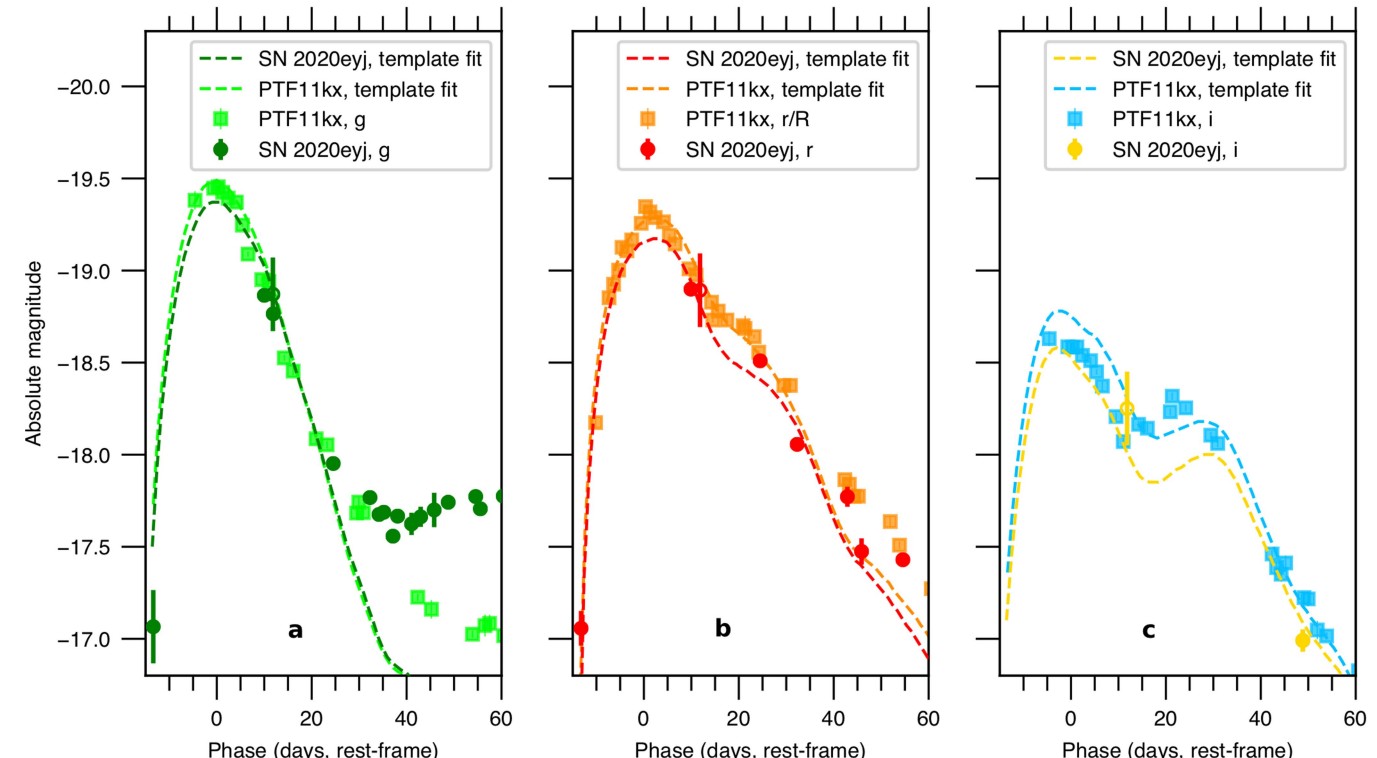

**Extended Data Fig. 1 | The light curves of SN 2020eyj are consistent with a SN Ia and its H-rich analogue PTF11kx.** We simultaneously fit the *g*, *r* and *i* light curves of the initial peak phases of both SN 2020eyj and PTF11kx with the SN Ia light curve fitter SNooPy. SN 2020eyj is well fit with stretch factor 1.2 and $E(B − V) = 0.5 ± 0.1$ mag. Similarly, PTF11kx is well fit with stretch factor 1.2 and $E(B − V) = 0.25 ± 0.02$ mag. Panels show the absolute-magnitude light curves of SN 2020eyj and PTF11kx, after correcting for the host extinction derived from the fit. **a**, *g* band. **b**, *r* band for SN 2020eyj and *r/R* band for PTF11kx. **c**, *i* band. Open circles indicate synthetic photometry derived from the spectra. The error bars represent 1σ uncertainties.

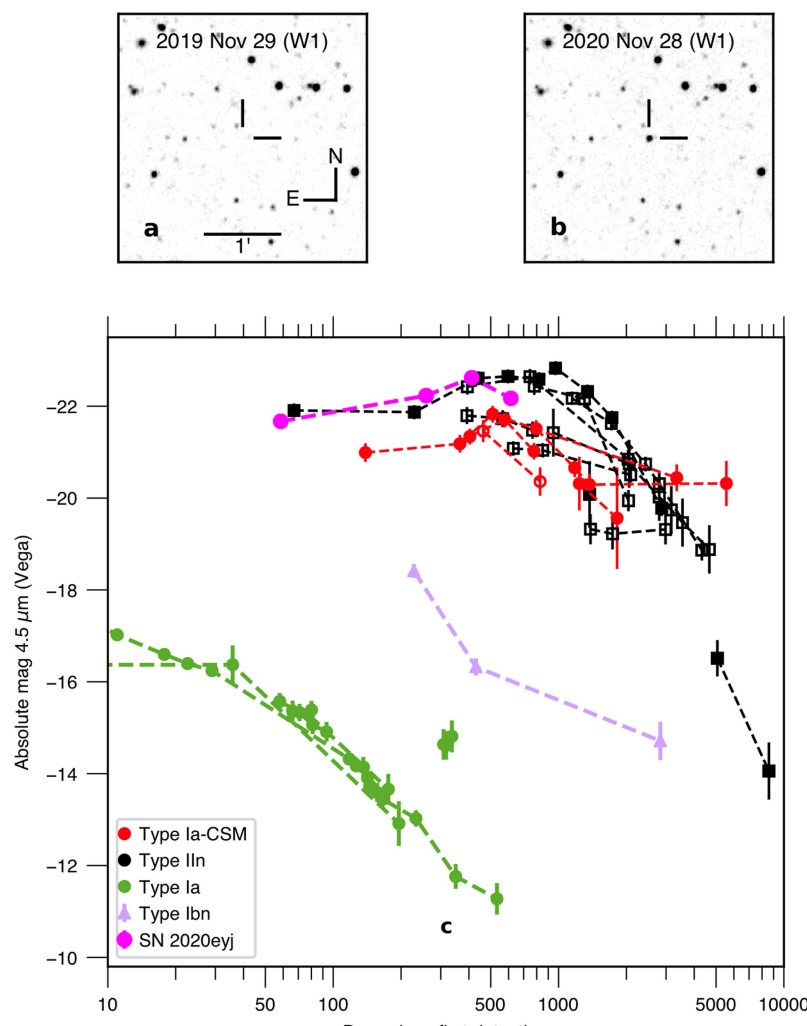

**Extended Data Fig. 2 | SN 2020eyj was accompanied by a bright mid-infrared counterpart. a**, A co-added image of the last NEOWISE-R epoch before the SN explosion, without any sign of the SN host. **b**, The co-added image in the W1 filter of the November 2020 NEOWISE-R epoch, 261 days after first detection, with SN 2020eyj clearly visible. **c**, A mid-infrared light-curve comparison of SN 2020eyj in the W2 filter (4.6 μm) to a sample of SNe observed with Spitzer at 4.5 μm, adapted from ref. 154, including Type IIn SNe (in black), Type Ia-CSM SNe (in red) and Type Ibn SN 2006jc (in lilac). Furthermore, the light curves of a sample of SNe Ia from ref. 155 is plotted in green. SN 2020eyj (large pink circles) is among the brightest SNe observed in the mid-infrared and is 6–10 magnitudes brighter than normal SNe Ia, depending on the phase. The error bars represent 1σ uncertainties.

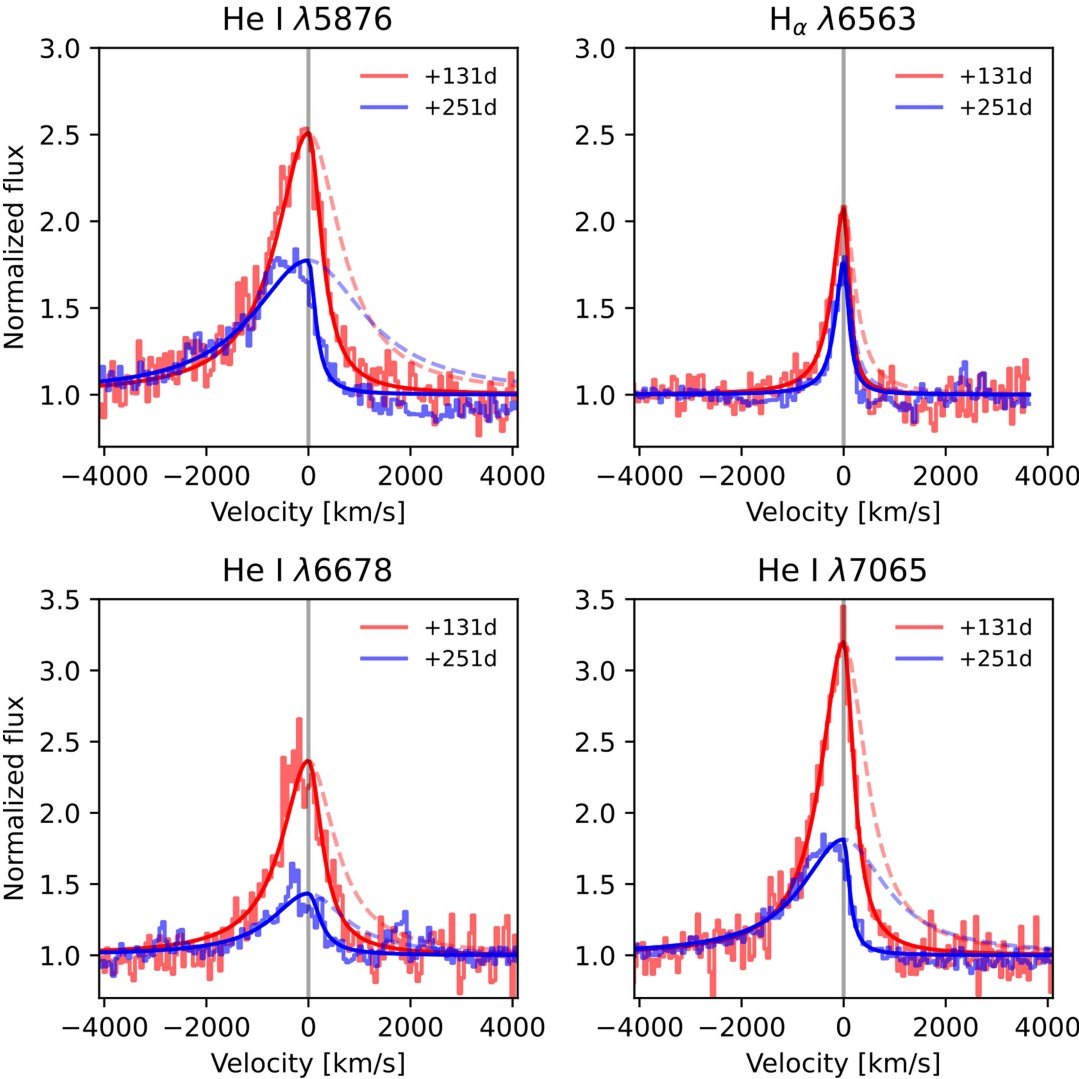

**Extended Data Fig. 3 | The He and Hα emission-line profiles in the late spectra of SN 2020eyj show notable asymmetry.** The He I emission lines at 5,876 Å, 6,678 Å and 7,065 Å all show strong attenuation in the red wings and an apparent blue shift over time between the 131 and 251 days epochs. Such line asymmetry is commonly observed in SNe Ia-CSM (ref. 11) and is interpreted as resulting from the condensation of dust in the ejecta or shocked CSM, obscuring the red wing (see 'Dust properties' section in Methods), but may also be a result of optical depth effects[20]. The Hα emission line at 131 days also shows asymmetry and there is a (minor) decline in flux between the two epochs shown here. By 329 days, the Hα luminosity has dropped to the level of the line emission in the host spectrum (see 'Optical spectroscopy' section in Methods).

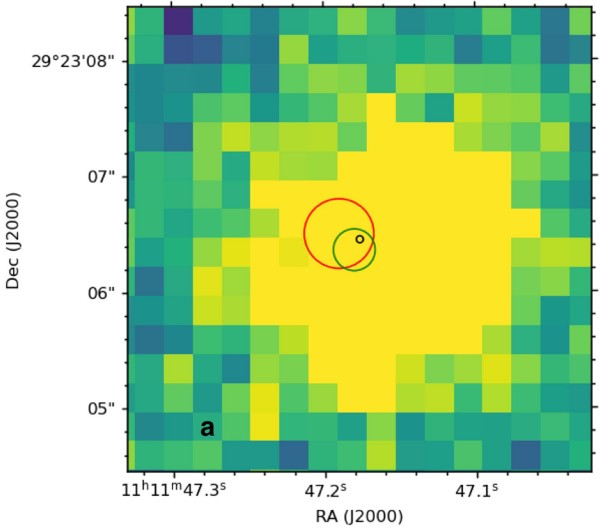

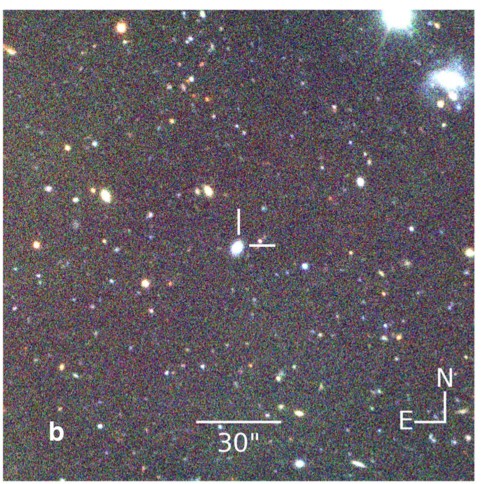

**Extended Data Fig. 4 | The position of the radio detection is consistent with the position of SN 2020eyj in the optical. a**, The average position of the e-MERLIN detections (black circle, 0.01″ uncertainty), the position reported in GaiaAlerts (*G* band, green circle, 0.06″ uncertainty) and the position of SN 2020eyj in the ALFOSC epoch at 382 days (*r* band, red circle, 0.1″ uncertainty), overlaid on a 4″ × 4″ Pan-STARRS1 *i*-band dataset of the host. **b**, A 3′ × 3′ colour composite image, obtained with NOT/ALFOSC, of the compact star-forming host galaxy of SN 2020eyj and its environment.

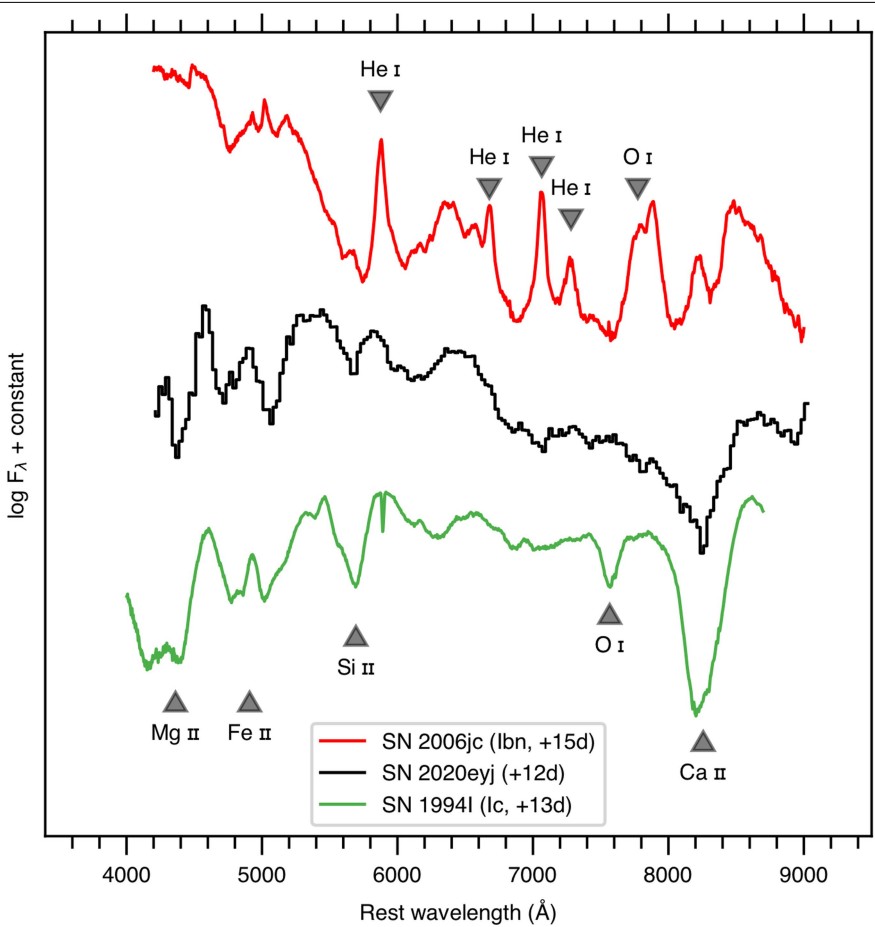

**Extended Data Fig. 5 | The spectra of SN 2020eyj do not match Type Ic or Ibn SNe at early epochs.** The SEDM classification spectrum of SN 2020eyj compared with Type Ibn SN 2006jc and Type Ic SN1994I at similar epochs, about 12 days after peak. At 15 days post-peak, SN 2006jc already showed strong He emission lines and developed the quasi-continuum typical for CSM-interaction-dominated spectra. These features are not observed in SN 2020eyj at 12 days post-peak but do become prominent at late phases (Fig. 2). At 13 days post-peak, SN 1994I has grown redder compared with its peak spectrum shown in Fig. 1 and the O I 7,774 Å absorption feature has become more prominent, whereas in SN 2020eyj, this feature is not present.

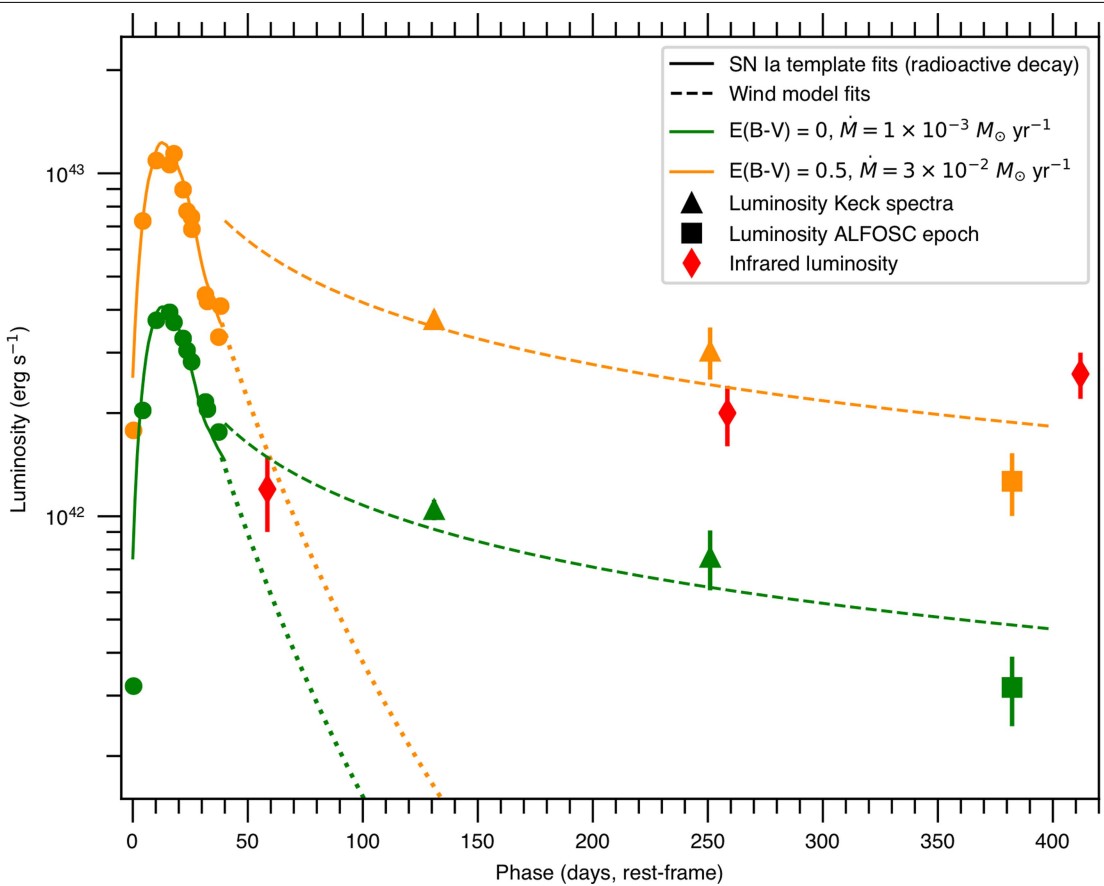

**Extended Data Fig. 6 | The bolometric light curve of SN 2020eyj can be described with a radioactive decay model for the peak phase and an optically thick wind for the tail phase.** For the initial SN Ia peak of SN 2020eyj, we adopt the bolometric light curve (solid lines) accompanying the SN Ia template fit to the *gri* photometry (see 'Light-curve fits' section in Methods), assuming no line-of-sight extinction (in green) and an extinction of $E(B − V) = 0.5$ mag (in orange). Overplotted are the associated bolometrically corrected luminosities up to 40 days. From epoch 46 days onward, the SN Ia template fit no longer accurately describes the observed (*g*-band) photometry (Extended Data Fig. 1). The dotted lines show the continuation of the bolometric light curve of the underlying SN Ia. The three measurements in the tail phase

are based on the integration of the two Keck spectra, extrapolated to the UV, and a bolometrically corrected photometric ALFOSC epoch. The dashed lines represent the fits to the tail-phase measurements using the analytical model from ref. 26, following the same colour scheme for the level of extinction. In the transition region from the diffusion peak to the CSM-interaction-powered tail, between 50 and 100 days, the sum of the models would overestimate the luminosity, suggesting that the CSM configuration is more complicated than a simple wind-like density profile. The red diamonds show the infrared luminosity of SN 2020eyj (Extended Data Table 3), which are not included in the model fits.

**Extended Data Table 1 | Key characteristics of SN 2020eyj**

| | |
|---|---|
| $\alpha$ (J2000) | $11^h 11^m 47.19^s$ |
| $\delta$ (J2000) | $+29°23'06.5''$ |
| Luminosity distance (Mpc) | 131.4 |
| First detection epoch (MJD) | 58915.212 |
| Peak epoch (fit, MJD) | $58929 \pm 2$ |
| Redshift | $0.0297 \pm 0.0001$ |
| $E(B{-}V)_{\mathrm{MW}}$ (mag) | 0.024 |

**Extended Data Table 2 | Log of spectroscopic observations of SN 2020eyj and its host and FWHM velocity measurements of prominent emission lines in the 131 and 251 days Keck spectra**

| MJD | Date (UT) | Phase (rest-frame) | Telescope + Instrument |
|---|---|---|---|
| 58941.2 | 2020 Apr 02 | 25 | P60+SEDM |
| 59050.3 | 2020 Jul 20 | 131 | Keck1+LRIS |
| 59150.5 | 2020 Oct 28 | 228 | P60+SEDM |
| 59162.2 | 2020 Nov 09 | 240 | NOT+ALFOSC |
| 59173.6 | 2020 Nov 30 | 251 | Keck1+LRIS |
| 59253.9 | 2021 Feb 08 | 329 | NOT+ALFOSC |
| 59615.4 | 2022 Feb 05 | 678 | Keck1+LRIS |

| | Line velocities | | | |
|---|---|---|---|---|
| | He I $\lambda 5876$ (km s$^{-1}$) | He I $\lambda 6678$ (km s$^{-1}$) | He I $\lambda 7065$ (km s$^{-1}$) | H$\alpha$ (km s$^{-1}$) |
| **131 days** | | | | |
| Full | 1080 | 960 | 780 | 400 |
| Blue wing | 1540 | 1270 | 1080 | 540 |
| **251 days** | | | | |
| Full | 1500 | 1150 | 1140 | 320 |
| Blue wing | 2680 | 1750 | 2000 | 390 |

Phase is relative to the first detection epoch, in the rest frame. The listed velocities are both the FWHM velocities measured from fitting the full line with a Lorentzian line profile as well as twice the half width at half maximum of the blue wing. The latter measurements better represent the true FWHM velocity, because the red wings in the emission lines are strongly attenuated (Extended Data Fig. 3).

**Extended Data Table 3 | Mid-infrared photometry from the WISE telescope and infrared and dust properties of SN 2020eyj**

|  |  | Epoch 1 | Epoch 2 | Epoch 3 | Epoch 4 |
|---|---|---|---|---|---|
| MJD |  | 58975.45 | 59181.42 | 59339.60 | 59548.01 |
| Phase | (rest-frame days) | 58.5 | 258.5 | 412.0 | 614.4 |
| W1 | (mag) | $17.29 \pm 0.03$ | $16.72 \pm 0.02$ | $16.40 \pm 0.02$ | $17.30 \pm 0.04$ |
| W2 | (mag) | $17.26 \pm 0.03$ | $16.70 \pm 0.04$ | $16.31 \pm 0.02$ | $16.76 \pm 0.03$ |
| $M_{\mathrm{dust}}$ | ($10^{-3}\ M_\odot$) | $1.8 \pm 0.3$ | $2.8 \pm 0.5$ | $4.8 \pm 0.5$ | $9.9 \pm 2.1$ |
| $T_{\mathrm{dust}}$ | (K) | $801 \pm 23$ | $809 \pm 27$ | $778 \pm 16$ | $608 \pm 23$ |
| $T_{\mathrm{BB}}$ | (K) | $1268 \pm 63$ | $1291 \pm 27$ | $1201 \pm 38$ | $826 \pm 35$ |
| $r_{\mathrm{BB}}$ | ($10^{16}$ cm) | $2.5 \pm 0.2$ | $3.2 \pm 0.3$ | $4.2 \pm 0.2$ | $6.4 \pm 0.6$ |
| $L_{\mathrm{BB}}$ | ($10^{42}$ erg s$^{-1}$) | $1.2 \pm 0.3$ | $2.0 \pm 0.4$ | $2.6 \pm 0.4$ | $1.4 \pm 0.3$ |
| Cumulative | ($10^{49}$ erg) | – | $2.7 \pm 0.4$ | $5.8 \pm 0.7$ | $9.3 \pm 1.0$ |

Phase (row 2, rest-frame days since first detection), magnitudes (rows 3 and 4, in AB system and binned per epoch), dust mass and temperature (rows 5 and 6, assuming 0.1 μm amorphous carbon grains), blackbody temperature, radius and luminosity (rows 7–9) and the cumulative radiated energy (row 10).

**Extended Data Table 4 | Host-galaxy photometry**

| Survey/Telescope | Instrument | Filter | Magnitude |
|---|---|---|---|
| GALEX | | $FUV$ | $21.37 \pm 0.38$ |
| GALEX | | $NUV$ | $20.99 \pm 0.14$ |
| Swift | UVOT | $w2$ | $21.45 \pm 0.16$ |
| Swift | UVOT | $m2$ | $21.66 \pm 0.27$ |
| Swift | UVOT | $w1$ | $21.31 \pm 0.24$ |
| SDSS | | $u$ | $21.16 \pm 0.23$ |
| SDSS | | $g$ | $20.26 \pm 0.06$ |
| SDSS | | $r$ | $19.86 \pm 0.05$ |
| SDSS | | $i$ | $19.74 \pm 0.13$ |
| PanSTARRS | | $g$ | $20.46 \pm 0.10$ |
| PanSTARRS | | $i$ | $19.94 \pm 0.08$ |
| PanSTARRS | | $z$ | $19.71 \pm 0.16$ |
| PanSTARRS | | $y$ | $20.21 \pm 0.57$ |
| NOT | ALFOSC | $r$ | $19.99 \pm 0.03$ |

Magnitudes are not corrected for reddening.