## [Peer Review File · Nature]

Manuscript Title: A radio-detected Type Ia supernova with helium-rich circumstellar material

Reviewer Comments & Author Rebuttals

Reviewer Reports on the Initial Version:

Referees' comments:

Referee #1 (Remarks to the Author):

This paper presents the first likely observation of a Type Ia supernova with a helium CSM, detectable in the radio. Thus it is very similar to the 2003 Nature paper by Mario Hamuy et al. on SN 2002ic with the first evidence of Hydrogen in a SN Ia.

However, as this is helium *and* it has a radio detection, its impact can be even more profound.

It hinges on the first figure, the comparison to other SNe Ia and the likelihood that this is a SN Ia - or at the very least a some type of thermonuclear supernova. Unfortunately the first spectrum is too late to see S II, the smoking gun for a SN Ia. However, the comparisons to the Ia-CSM and 91T-like supernovae at this phase, and the difference between that and the Ic (SN 1994I) is convincing enough for me. I think some readers though would like to see more comparisons to SNe Ic at phases from +7 to +21 days - matching the lightcurve of this event. I think it would also be useful to show a comparison to a Ibn at this phase, to highlight the differences even though much later on they look similar.

The one thing I take issue with in the paper is the discussion of the extinction. I am glad the authors show two possibilities in Figure 8, but I think applying this to just the tail is naive. The assumption that the colors of this event are normal in any way, shape, or form is a bit hard to believe, thus the fact that the SN is quite red at early times may just be a property of such an event, and thus applying a color correction via SNOOPy fits may be completely erroneous.

Is there any evidence from the presence of Na ID lines in the SN spectra that this SN has extinction? Is there evidence that the Na ID lines were there and vanished? If not, then I think an analysis assuming 0 extinction for the entire lightcurve must be presented as well. When you find a rare event, you can hardly assume that under all this CSM it is normal...

Referee #2 (Remarks to the Author):

Dear Leslie Sage,

Please find below my report on the paper entitled "A radio-detected thermonuclear supernova from

a

single-degenerate progenitor with a helium star donor" by Kool and collaborators. The paper presents an impressive set of observations for a Type Ia Supernova with clear signs of interaction starting about a month after maximum light. Evidence for interaction comes from independent sources including the persisting peculiarities of the nebular optical spectra (narrow emission lines, strong FeII emission as observed in SNe Ia-CSM), the high mid-IR luminosity (comparable to that observed in super-luminous SNe IIn), and the unprecedented detection of radio emission. All these constraints give credence to the single-degenerate scenario involving a white dwarf and a He-star companion, and therefore represents an important result for the community since such constraints are very rarely obtained. The paper is well written and comprehensive.

I have no major criticism of this work but provide some suggestions for improvement and clarifications.

Best regards, the referee

=====

Primary points:

The authors emphasize the distinction between the single-degenerate (SD) and the double-degenerate (DD) scenario. Equally important for combustion physics and the origin of Type Ia SNe is to understand whether the exploding white dwarf is at the Chandrasekhar mass or below. A mention of this aspect should be made in the introduction. The present observations might set some constraints on the white-dwarf mass. For example, is the lack of OI7774 suggestive of a sub-M_{ch} white dwarf? Or the short rise time?

In the context of the single-degenerate scenario with a He-star companion, some He-rich material should be stripped from the companion by the ejecta. This material would then be present in the inner, slow ejecta and be observable as narrow emission lines when the ejecta turns optically thin after maximum (e.g., see Liu+12, Botyanszki+18, Dessart+20). This line emission would stem from radioactive decay and thus show an exponential decline in time, with a total line power limited by the absorbed decay power (Dessart+20). The HeI lines observed in SN2020eyj (time of appearance, strengths, evolution in time) seem to contradict this scenario but it would be good if the authors could demonstrate it. This would give further support to the interaction scenario. Alternatively, could HeI lines arise from stripped material in the inner ejecta while interaction takes place in the outer ejecta?

Using the present observations of SN2020eyj, the authors promote the SD scenario involving a white dwarf and a He star. How does this model fit within the overall SD scenario and the possibility of having an H-rich star as a companion. Should such systems with a H-rich star outnumber the systems with a He-star companion? Please briefly address this aspect.

The brightness of the plateau that starts at 50d post-discovery is small relative to the main peak so it is not clear that the interaction did not start much earlier but was simply dominated by the much stronger contribution from decay power. In relation to this, can we firmly exclude the presence of

weak narrow HeI lines in the first spectrum or the spectrum is too low quality for this verification -- an early detection of narrow HeI lines would be clear evidence that such lines form in the outer ejecta, hence in an interaction, rather than in the inner ejecta (in association with stripped material). Also, the adoption of 10000km/s for the SN Ia ejecta velocity seems very small. This seems to be a standard photospheric velocity around max (of a velocity at maximum absorption in some line) but there is material at larger velocity (up to ~40000km/s?), with large specific kinetic energy, and with strong potential impact (the low optical depth of this outer material is irrelevant).

Mid-infrared emission: The curves shown in Fig 5 and the origin of this mid-IR emission in interacting supernova is perplexing. First, the selection of events in Fig 5 is misleading. The point of the figure would be to distinguish interacting (luminous mid-IR) and non-interacting (faint mid-IR) events so mixing events the way the authors did is not ideal. The authors indicate the mid-IR luminosities for SNe Ia and they should do the same with representative, non-interacting SNe Ibc. SN2014C is an exceptional event that morphed from Ib to IIc and thus not a good non-interacting Ib (SN2014C shares many similarities with SN2020eyj and should be briefly discussed -- the interaction configuration may be analogous). The authors write that SN2020eyj is 2-4 magnitude brighter than SNe Ia and Ibc, but in fact it is more like 10 magnitudes brighter than non-interacting counterparts (i.e., if we exclude 2014C etc). There is also a large offset in optical brightness too, which is more like 2-4 magnitudes. Hence, the interaction likely boosts the brightness in all bands (i.e., a bolometric boost), but there is an additional boost in the mid-IR. The knowledge of a -22mag tells one nothing of the actual erg/s coming out in the mid-IR relative to the optical so connecting to energy or power would be useful to understand this SN (i.e., by quantifying L_{opt} , L_{IR} , L_{radio} etc at various epochs and their integral over time if possible).

=====

Minor points (roughly in order of appearance):

In most figures, the legend overlaps with the data or the model curves, which should be corrected.

The historical scenario that SNe Ibc presumably arise from massive Wolf-Rayet stars is in tension with the strength of HeI lines (see recent study of Dessart+22). This should be updated.

He CSM -> He-rich CSM (throughout)

One weakness of the paper is the terse discussion of the SN Ia associated with SN2020eyj. How standard was SN2020eyj in terms of bolometric luminosity, rise time, fading rate, etc. If normal looking, how can we reconcile that with the exceptional presence of HeI lines and radio emission at late times? The authors say in the main text that "it is unclear how representative SN2020eyj is". However, they provide some clues in the Methods section, which could appear as an informative summary in the main text.

What is the fractional power emitted in the radio at late times (relative to L_{bol} or $L_{optical}$), assuming plausible spectral energy distributions or slopes in the radio?

Replace He emission lines and similar instances with HeI emission lines (same for Fe, which is typically FeII here).

In all figures, make sure tick marks are present at bottom and top of plots. Use major as well as minor tick marks. It is hard to interpret figures without these.

Times are confusing (even if the nomenclature is defined early on). The authors use +N days for N days post discovery but this usually refers to days post bolometric or B-band maximum. To avoid confusion, I would suggest always dropping the "+" and just state pre or post-discovery, or pre- or post-maximum etc.

Methods:

p27: "quasi-continuum" is awkward when referring to the emission blueward of 5700Å and interpreted as arising from FeII lines rather than continuum emission. Some reference would be good too. Referring to FeII line emission (or blue quasi-featureless emission) would be more adequate.

"receding red wing" : awkward.

There is discussion on reddening in various parts of the paper. A small MW $E(B-V)$ is given on page 28, but that is much smaller than the host reddening with $E(B-V)$ of ~ 0.5 mag. In some figures, it is not said whether correction for reddening has been made. So, it would help grouping the key observational characteristics of the SN early in the paper and stating clearly in the captions what is shown.

Fig 2: Please add a normal SN Ia at late times to show how much it differs. It is interesting that OI774 is absent but that MgII emission probably broadens the 8500Å emission feature (O and Mg are usually present simultaneously in stars so having one without the other is strange). Also, the caption should probably report the striking asymmetry of the H α and HeI line profiles, absent in 06jc early on (but present at similar late times?).

line 786: is it not surprising that the g-r color does not change much over the span 100-400d despite the presence of dust. How does this compare to the Type Ibn SN2006jc? Is that indicative that the dust is external?

line 819: "fast for a SN Ia but similar to 11kx": Is that indicative of a sub-Mch white dwarf? This seems very important in the context of this work. There is abundant literature on the rise times of sub-Mch and Mch explosion models, which could be used to discuss this observational characteristics.

line 910: what are the quoted errors on the dust mass? The uncertainty is probably much larger (order ten?), because of the uncertainty on the physics of dust (absorption, scattering, dust composition, clumping, location, extent, etc). The authors conclude that the mid-IR emission is dominated by pre-existing dust. which then implies that the huge mid-IR magnitude as early as

59days (which is nearly the max) is largely driven by the huge flux from the SN and the interaction. The similarity of 2020eyj with other SNe IIn and Ia-CSM suggest that pre-existing dust may be common in those events and thus suggest that the SN itself is not the agent producing the dust, in conflict with many recent claims (for example for SN2010jl; Gall+14). Are SNe strong dust producers?

line 930: If the dust is pre-existing and external, it cannot cause an asymmetry in the HeI line profiles (HeI photons emitted in the interaction region cross essentially the same column of dust on their way out).

Do the authors imply that there would be both external dust as well as dust forming in the interaction regions where the HeI lines form? Blue-red asymmetry of HeI line profiles are seen in radiative-transfer calculations of SNe Ibn (see, for example, Dessart+22) and suggest that standard optical depth effects can produce profile asymmetry in 1D with no dust. This option should be mentioned.

In the section on the CSM origin, CSM masses up to $\sim 1M_{\text{sun}}$ are quoted. Such a mass is comparable to the mass of the exploding star. Should this not lead to a major interaction and a greater radiative power?

Referee #3 (Remarks to the Author):

1. Summary of the key results

The paper reports the first detection ever of a Type Ia Supernova (SNIa) at radio wavelengths.

SNIa exploding in vacuum do not emit in the radio domain. Radio emission from these explosions is expected only if the material they eject (a.k.a. ejecta) crashes into circumstellar matter (CSM) lost by the progenitor prior to the explosion, and placed at relatively close distances from the explosion site. Since only some progenitor channels are expected to produce such a configuration, a radio detection strongly constrains the nature of the system at the origin of the explosion.

From the spectroscopic point of view, the early data show that the SN does belong to the Ia class (sub-class 1991T-like). At later epochs, narrow emission lines appear in the spectra, marking the onset of a delayed ejecta-CSM interaction. From the weakness of the Hydrogen lines and the strength of the Helium lines, the authors deduce that the CSM is He-rich.

Based on these findings, the authors claim that the progenitor was a single-degenerate binary system in which a white dwarf accreted material from a helium star donor.

2. Originality and significance

The work presented by the authors is not original in terms of techniques and context, in the sense that this field has been investigated for a long time. However, all radio searches so far have failed to detect any Ia, even in cases in which other signs of the possible presence of CSM were present in

the spectra and/or in the light curves of some of these objects. The lack of a radio detection is one of the arguments used to disfavor the so-called single-degenerate (SD) scenario because, especially in the case of a red-giant donor, significant amounts of CSM would be expected. The same argument was used to favor the double-degenerate (DD) scenario, which is expected to have a “clean” environment at the time of the explosion. Truly enough, some authors maintain that ejecta-CSM interaction could take place also in the DD case, if the common-envelope lost by the system prior to the binary merger is not too far from the explosion. But this requires quite some fine tuning.

At face value, this is the only SNIa that has produced a detectable radio emission (despite of the extended searches). The weakness of H lines and the strength of the He lines is also a unique finding, so far. Both aspects seem to suggest that these events are rare and can hardly explain the bulk of SNIa. For this reason the findings reported by this paper do not answer the long-standing question about the progenitor’s nature, hence casting some doubts on the general significance of the results.

Nevertheless, the paper presents the first direct evidence that a single-degenerate system hosting a He-donor can produce a SN Ia explosion.

What fraction of the observed overall SNIa rates this channel can explain remains to be seen and other aspects also need to be clarified (effects of the timing of radio observations, distribution and distance of the CSM material, presence/absence of cavities, ...). However, the result is unprecedented and significant in itself. In addition to the pure detection (which is worth in itself), the radio flux provides additional and independent information, which permits to retrieve some important physical parameters.

3. Data & methodology

The data quality is high and definitely suited for the study which is presented. The methodology is standard, free of exotic procedures and/or obscure statistical analysis. There is absolutely no concern about the signatures which lead the authors to their conclusions.

4. Appropriate use of statistics and treatment of uncertainties

Error bars are properly computed and reported. In general the signal-to-noise is very high for all the data used to draw the conclusions.

5. Conclusions

The paper reports quite a clear-cut case: a SN Ia (belonging with rather high confidence to the high luminosity tail of the so called Branch-normal Type Ia, a.k.a. 91T-like) shows undoubtable signs of interaction with He-rich material. This was the long-sought smoking gun that would speak in favour of the so-called single-degenerate channel, in which a C-O white dwarf accretes from some non-degenerate star. The case presented by the authors is very favorable, and leaves little doubt. In other cases showing traces of interaction in the form of narrow emission lines, this had already started at very early epochs, hence making it impossible to get a “clean” signal that would allow to establish the exact nature of the explosion. The delayed onset of the interaction (explained as a cavity surrounding the progenitor, a scenario that has been invoked in the past in this context to

account for the lack of radio emission) allowed the authors to properly classify the object as a SNIa. On top of this, and for the first time, as opposed to the known cases of interacting Ia (SNe Ia-CSM), the material appears to be H-poor and He-rich, which gives extra information about the donor star.

The evidences provided by the authors are solid, and do not depend very much on models or complex interpretations:

- 1) The SN is of the Ia type
- 2) It shows signs of interaction with H-poor, He-rich material
- 3) It shows the first ever radio detection

The conclusions reached by the authors are quite robust and do not leave much room for speculations.

6. Suggested improvements

Here follow some minor comments and suggestions the authors should consider. The comments refer to the indicated manuscript line/s.

24 – “from a companion star”, reference 1: I am not sure this is the appropriate reference, as the original suggestion came much earlier than 2011.

35-38 : Although it is understood that the space is limited, the authors should also briefly mention the search for circumstellar material placed at distances sufficiently large that no interaction (and hence no radio emission) is expected. The work done on the search for narrow absorption lines should be cited (Patat+ 2007, Simon+ 2007, Maguire+ 2014, Sternberg+ 2014). Incidentally, the first paper introduced the possibility that recurrent pre-explosion nova outbursts could excavate the CSM, hence weakening the link between lack of radio detections and DD progenitors).

62: reference 30. This refers to a paper that deals with the same topic, in general terms. However, that article reports the detection of narrow absorption lines from neutral CSM, and has nothing to do with prominent Balmer emission lines. Although the paper should be referred to in the general introduction (see the comment above), the reference here is not appropriate.

106: “Although the thermonuclear nature of Type Iax SNe is debated” : It would be useful to add a reference in support of this statement.

150-151: Although interesting, this sentence is not substantiated by quantitative arguments. One would expect that the circumstellar disk is comparatively small, and hence it should be quickly overtaken by the SN explosion, which would most likely also evaporate the dust. If that is the case, it is hard to identify this dust as the source of the IR emission from the SN. The authors should briefly discuss this.

167-170: The authors state that the He star+WD is likely the dominant source of short-delay SNIa.

However, no reference is given in support of this statement. Also, there is no context for the reader to judge how important this contribution would be, for instance, in terms of rates.

176-177: The authors state that this is the first SN Ia where a SD origin is confirmed through a radio detection. The presence of a dense and detached CSM is witnessed by the light curve and the narrow emission lines in the spectra. The nature of the donor is deduced from the strength of He lines and the weakness of the H lines. Although the radio detection is certainly important, it is not clear why it should confirm the SD nature of the progenitor. This can probably be fixed by replacing “confirmed through” with “supported by”.

7. References

The references are quite complete given the limited amount of space. See the notes in the suggested improvements about a set of references which should be added to complete the information about the context.

8. Clarity and context

The paper provides a sufficient context for non-experts. The text presents the case effectively and clearly, and it reads very well. The detailed supplementary material is well presented and provides the expert reader with sufficient information to judge the quality of the data and their analysis.

Author Rebuttals to Initial Comments:

2 Referee 1

2.1 ✓ Comment 1

I think some readers though would like to see more comparisons to SNe Ic at phases from +7 to +21 days - matching the lightcurve of this event. I think it would also be useful to show a comparison to a Ibn at this phase, to highlight the differences even though much later on they look similar.

Reply to the referee: We thank the referee for this suggestions, and have added a figure (see page 53) to Methods that compares the classification spectrum of SN 2020eyj to the Type Ic SN 1994I and Type Ibn SN 2006jc at a similar phase of 12 days post peak.

2.2 ✓ Comment 2

The one thing I take issue with in the paper is the discussion of the extinction. I am glad the authors show two possibilities in Figure 8, but I think applying this to just the tail is naive. The assumption that the colors of this event are normal in any way, shape, or form is a bit hard to believe, thus the fact that the SN is quite red at early times may just be a property of such an event, and thus applying a color correction via SNooPy fits may be completely erroneous. Is there any evidence from the presence of Na ID lines in the SN spectra that this SN has extinction? Is there evidence that the Na ID lines were there and vanished? If not, then I think an analysis assuming 0 extinction for the entire lightcurve must be presented as well. When you find a rare event, you can hardly assume that under all this CSM it is normal.

Reply to the referee: This is a good point, and we have updated the figure to include an observed peak bolometric light curve, without assuming extinction (see page 54). Regarding Na ID absorption, the peak spectrum does not have sufficient spectral resolution to determine this feature, and in the late spectra it is not possible to determine this feature at a reasonable level of significance, because the SN is much fainter and the absorption line falls in the (asymmetric) red wing of a prominent He I emission line.

3 Referee 2

3.1 ✓ Comment 1

The authors emphasize the distinction between the single-degenerate (SD) and the double-degenerate (DD) scenario. Equally important for combustion physics and the origin of Type Ia SNe is to understand whether the exploding white dwarf is at the Chandrasekhar mass or below. A mention of this aspect should be made in the introduction. The present observations might set some constraints on the white-dwarf mass. For example, is the lack of OI7774 suggestive of a sub-M_{ch} white dwarf? Or the short rise time?

Reply to the referee: It is a valid point that the distinction between Chandrasekhar and sub-Chandra is important to the origin of SNe Ia in general, and a peculiar SN Ia like SN 2020eyj (and PTF11kx) in particular. However, this is a question that has a lot of nuance to it, and space is limited in the paper. The literature on sub-Chandra models are typically geared toward explaining normal SNe Ia, and there are no clear predictions of these types of events. Also, the fact that two components, the SN Ia and CSM interaction, contribute to the light curve of SN 2020eyj at a ratio that changes with time, makes it difficult to put solid constraints on what the underlying SN Ia looks like. This in turn makes it difficult to distill what properties of SN 2020eyj could be interpreted in the context of sub-Chandra SNe Ia. Of course, this uncertainty of what SN 2020eyj intrinsically look like as a Ia, is also addressed in other comments from the referees, so we have made changes to the paper to better reflect this uncertainty. Also, we now highlight how the observed properties in which SN 2020eyj (and PTF11kx) differ from the normal SN Ia population, such as rise time and peak spectrum properties, may be leveraged to discover and study more such events (see page 4/5). In time, when we increase the sample of such delayed interaction SNe Ia and the unusual properties persist for these events, the discussion regarding Chandrasekhar vs sub-Chandra SN Ia should definitely be held.

3.2 ✓ Comment 2

In the context of the single-degenerate scenario with a He-star companion, some He-rich material should be stripped from the companion by the ejecta. This material would then be present in the inner, slow ejecta and be observable as narrow emission lines when the ejecta turns optically thin after maximum (e.g., see Liu+12, Botyanszki+18, Dessart+20). This line emission would stem from radioactive decay and thus show an exponential decline in time, with a total line power limited by the absorbed decay power (Dessart+20). The HeI lines observed in SN2020eyj (time of appearance, strengths, evolution in time) seem to contradict this scenario but it would be good if the authors could demonstrate it. This would give further support to the interaction scenario. Alternatively, could HeI lines arise from stripped material in the inner ejecta while interaction

takes place in the outer ejecta?

Reply to the referee: While the detection of stripped material would certainly be intriguing, there is no reason to believe this is the case with SN 2020ejj. The late spectra of SN 2020ejj evolve very much like those of SNe Ibn, with HeI emission lines growing broader over time and CSM-interaction driven features dominating the spectra until the SN fades from view, obscuring the inner ejecta. Also, for the stripped material you would expect FWHM velocities <1000 km/s (Pan+2010,2012), and there is no sign for such a narrow component in the spectra. We now mention this in the text (see page 20/21).

3.3 ✓ Comment 3

Using the present observations of SN2020ejj, the authors promote the SD scenario involving a white dwarf and a He star. How does this model fit within the overall SD scenario and the possibility of having an H-rich star as a companion. Should such systems with a H-rich star outnumber the systems with a He-star companion? Please briefly address this aspect.

Reply to the referee: According to Ruiter+09, it is estimated $\sim 10\%$ of SD SNe Ia arise from the He + WD channel. We have now included this in the main text (see page 5).

3.4 ✓ Comment 4

The brightness of the plateau that starts at 50d post-discovery is small relative to the main peak so it is not clear that the interaction did not start much earlier but was simply dominated by the much stronger contribution from decay power. In relation to this, can we firmly exclude the presence of weak narrow HeI lines in the first spectrum or the spectrum is too low quality for this verification – an early detection of narrow HeI lines would be clear evidence that such lines form in the outer ejecta, hence in an interaction, rather than in the inner ejecta (in association with stripped material). Also, the adoption of 10000km/s for the SN Ia ejecta velocity seems very small. This seems to be a standard photospheric velocity around max (of a velocity at maximum absorption in some line) but there is material at larger velocity (up to 40000km/s?), with large specific kinetic energy, and with strong potential impact (the low optical depth of this outer material is irrelevant).

Reply to the referee: It is true we cannot firmly exclude a contribution by CSM interaction to the main peak, but SN 2020ejj is relatively faint compared to other SNe Ia-CSM and shows a similar light curve evolution to PTF11kx, which makes us believe this contribution is likely to be minor. The first spectrum unfortunately does not have the spectral resolution to exclude the presence of narrow emission lines, but it clearly lacks any other sign of CSM interaction which we would expect from a SN Ibn at this phase, as made clear by the

comparison with SN 2006jc at a similar phase in the new Figure 5 (see page 53).

With regards to the ejecta velocity, we adopted 10.000 km/s as it is a commonly used assumption for SNe Ia, we have added a reference (Wilk+18) (see page 2). If there is sufficient material at larger velocities such that its interaction with the CSM contributes significantly to the light curve, this would affect the size of the CSM-free (or CSM-poor) inner cavity to the extent it would be larger by a factor of a couple than what we present in the paper. Following Wilk+18, little ejecta is accelerated above 25.000 km s⁻¹.

3.5 ✓ Comment 5

Mid-infrared emission: The curves shown in Fig 5 and the origin of this mid-IR emission in interacting supernova is perplexing. First, the selection of events in Fig 5 is misleading. The point of the figure would be to distinguish interacting (luminous mid-IR) and non-interacting (faint mid-IR) events so mixing events the way the authors did is not ideal. The authors indicate the mid-IR luminosities for SNe Ia and they should do the same with representative, non-interacting SNe Ibc. SN2014C is an exceptional event that morphed from Ib to IIn and thus not a good non-interacting Ib (SN2014C shares many similarities with SN2020eyj and should be briefly discussed – the interaction configuration may be analogous). The authors write that SN2020eyj is 2-4 magnitude brighter than SNe Ia and Ibc, but in fact it is more like 10 magnitudes brighter than non-interacting counterparts (i.e., if we exclude 2014C etc). There is also a large offset in optical brightness too, which is more like 2-4 magnitudes. Hence, the interaction likely boosts the brightness in all bands (i.e., a bolometric boost), but there is an additional boost in the mid-IR. The knowledge of a -22mag tells one nothing of the actual erg/s coming out in the mid-IR relative to the optical so connecting to energy or power would be useful to understand this SN (i.e., by quantifying L_{opt} , L_{IR} , L_{radio} etc at various epochs and their integral over time if possible).

Reply to the referee: Like the referee points out, the intention of the figure is to highlight the IR brightness of SN 2020eyj, compared to normal non-interacting SNe. Unfortunately the number of SNe for which mid-IR coverage exists is very limited, in particular normal Type Ib/c, which is why we opted to include all SNe observed with the Spitzer space telescope. However, upon reflection we agree with the referee that this can be confusing, so we have updated the figure to only show the IR-bright interacting SNe (Ia-CSM, IIn, Ibn) and the sample of normal Ia SNe. We have updated the text in the IR section and the caption accordingly (see page 28/47).

Additionally, we have added the IR luminosity light curve of SN 2020eyj to the figure showing the fits to the bolometric light curve (see page 54), to the optical and IR components can be compared more easily. We do note that the IR component is not included in the model fits, since there is likely a time delay between the optical and IR emission. We also explicitly mention in the text

now that the integrated energy radiated by the optical and IR components is of similar order of magnitude.

Regarding SN 2014C, we thank the referee of reminding us of this SN. We have added a mention of it in the CSM shell discussion (see page 31).

3.6 ✓ Comment 6

One weakness of the paper is the terse discussion of the SN Ia associated with SN2020eyj. How standard was SN2020eyj in terms of bolometric luminosity, rise time, fading rate, etc. If normal looking, how can we reconcile that with the exceptional presence of HeI lines and radio emission at late times? The authors say in the main text that "it is unclear how representative SN2020eyj is". However, they provide some clues in the Methods section, which could appear as an informative summary in the main text.

Reply to the referee: We agree that it is worth highlighting in what way SN 2020eyj may be different from normal SNe Ia, because this would also aid the search for similar SNe. It is intriguing that both SN 2020eyj and PTF11kx have relatively fast rise times, so we address this now in the main text (see page 4/5). As to the peak magnitude and decline rate, we have tried to convey how SN 2020eyj can be reconciled with a normal SN Ia, but may have put too much emphasis here. We now added an observed bolometric luminosity light curve (see page 54), and make it clear in the text that light curve properties of SN 2020eyj could also be intrinsic, rather than (solely) due to dust extinction (see page 26).

3.7 ✓ Comment 7

What is the fractional power emitted in the radio at late times (relative to L_{bol} or L_{optical}), assuming plausible spectral energy distributions or slopes in the radio?

Reply to the referee: The late-time radio luminosity decreases roughly as t^{-1} with time after the peak, and with frequency as ν^{-1} . If we restrict at times of around 600-800 d, when our radio observations were taken, we can obtain a total radio power of $L_R \approx \nu L_{\text{nu}} = 5.1 \times 10^9 \text{ Hz} * (1 - 2) \times 10^{27} \text{ erg/s/Hz} = (0.5 - 1) \times 10^{37} \text{ erg/s}$. Even if we had assumed that the radio spectrum extended from, say 100 MHz up to 30 GHz, L_R is several orders of magnitude less than L_{bol} .

3.8 ✓ Comment 8

There is discussion on reddening in various parts of the paper. A small MW $E(B-V)$ is given on page 28, but that is much smaller than the host reddening with $E(B-V)$ of 0.5mag. In some figures, it is not said whether correction for reddening has been made. So, it would help grouping the key observational

Figure 1: Comparison of Type Ia SN 2004eo and SN 2020eyj at late phases

characteristics of the SN early in the paper and stating clearly in the captions what is shown.

Reply to the referee: Thanks for the suggestion, we have now added a table summarizing the properties of SN 2020eyj (see page 45). Also, where missing, we have added to the captions information about dereddening.

3.9 ✓ Comment 9

Fig 2: Please add a normal SN Ia at late times to show how much it differs. It is interesting that OI774 is absent but that MgII emission probably broadens the 8500Å emission feature (O and Mg are usually present simultaneously in stars so having one without the other is strange). Also, the caption should probably report the striking asymmetry of the H α and HeI line profiles, absent in 06jc early on (but present at similar late times?).

Reply to the referee: For reference, we have added a comparison figure to this report of SN 2020eyj and the normal Type Ia SN 2004eo, see below. As expected, they are quite dissimilar, but what this comparison doesn't convey is that at this late phase SN 2004eo is much fainter than SN 2020eyj, by 4 magnitudes. As such, we would not expect to observe any underlying SN Ia features. We have now noted this in the text (see page 24), but refrain from adding the SN 2004eo spectrum to the already busy Fig. 2. We have updated the caption of Fig. 2 to highlight the asymmetry in the emission features.

3.10 ✓ Comment 10

line 786: is it not surprising that the g-r color does not change much over the span 100-400d despite the presence of dust. How does this compare to the Type Ibn SN2006jc? Is that indicative that the dust is external?

Reply to the referee: That is correct, the lack of color evolution is consistent with pre-existing dust. From the IR properties, we conclude that most, if not all, of the IR flux is due to an IR echo from pre-existing dust rather than dust formation, and the lack of color evolution supports this. In the case of SN 2006jc, where new dust was formed, the light curves in the optical filters at shorter wavelengths declined faster, as expected from the increase of extinction and the extinction law. Thank you for bringing this to our attention, we have clarified this in the text (see page 30).

3.11 ✓ Comment 11

line 819: "fast for a SN Ia but similar to 11kx": Is that indicative of a sub-Mch white dwarf? This seems very important in the context of this work. There is abundant literature on the rise times of sub-Mch and Mch explosion models, which could be used to discuss this observational characteristics.

Reply to the referee: We agree it is interesting that SN 2020eyj and PTF11kx share this aspect, so we have put more emphasis on it in the main text (see page 4/5). However, to constrain the mass of the exploding SN Ia based on light curve properties, given that CSM interaction (which is not easily explained in the context of sub-Chandra events) plays a role too, we think we would be over-interpreting the data.

3.12 ✓ Comment 12

line 910: what are the quoted errors on the dust mass? The uncertainty is probably much larger (order ten?), because of the uncertainty on the physics of dust (absorption, scattering, dust composition, clumping, location, extent, etc). The authors conclude that the mid-IR emission is dominated by pre-existing dust. which then implies that the huge mid-IR magnitude as early as 59days (which is nearly the max) is largely driven by the huge flux from the SN and the interaction. The similarity of 2020eyj with other SNe IIn and Ia-CSM suggest that pre-existing dust may be common in those events and thus suggest that the SN itself is not the agent producing the dust, in conflict with many recent claims (for example for SN2010jl; Gall+14). Are SNe strong dust producers?

Reply to the referee: The quoted errors are the statistical errors from the fit. The dust mass depends on grain size, which we can not constrain based on the available data. The dust mass can vary up to an order of magnitude between grain sizes of 0.01 to 1 micron (we have assumed a grain size of 0.1

micron). We have now included an explicit mention of this in the text (see page 28).

The IR properties are consistent with pre-existing dust, and while a contribution from dust formation can not be ruled out, it is not necessary to invoke dust formation to explain the IR (in particular since the referee noted the line asymmetry could also arise from optical depth effects, see below). We have now emphasized this point in the paper (see page 30).

3.13 ✓ Comment 13

line 930: If the dust is pre-existing and external, it cannot cause an asymmetry in the HeI line profiles (HeI photons emitted in the interaction region cross essentially the same column of dust on their way out). Do the authors imply that there would be both external dust as well as dust forming in the interaction regions where the HeI lines form? Blue-red asymmetry of HeI line profiles are seen in radiative-transfer calculations of SNe Ibn (see, for example, Dessart+22) and suggest that standard optical depth effects can produce profile asymmetry in 1D with no dust. This option should be mentioned.

Reply to the referee: Yes, we suggest in the paper that the IR emission predominantly arise from an IR echo by pre-existing dust, but that based on the line profiles there is evidence for dust formation. It is now clear to us from the reference you provided that this is not necessarily the case, as the asymmetry could also be a result of optical depths effects, thank you for this. We have now included a mention of this option (see page 30).

3.14 ✓ Comment 14

In the section on the CSM origin, CSM masses up to 1Msun are quoted. Such a mass is comparable to the mass of the exploding star. Should this not lead to a major interaction and a greater radiative power?

Reply to the referee: The SN ejecta will be gradually decelerated by an extended CSM, and the conversion of kinetic energy depends on CSM properties such as the CSM density. If we look at the late spectra, the Fe II lines that make up the quasi-continuum must be broad because the continuum appears smooth. This implies the CSM does not decelerate the ejecta as much as in other interacting SNe (such as IIn), which would explain a relatively small radiate power.

3.15 ✓ Comment 15

In most figures, the legend overlaps with the data or the model curves, which should be corrected.

Reply to the referee: This has now been corrected.

3.16 ✓ Comment 16

The historical scenario that SNe Ibn presumably arise from massive Wolf-Rayet stars is in tension with the strength of HeI lines (see recent study of Dessart+22). This should be updated.

Reply to the referee: The text has been updated to reflect the study by Dessart et al (see page 3).

3.17 ✓ Comment 17

He CSM to He-rich CSM (throughout)

Reply to the referee: This has now been fixed.

3.18 ✓ Comment 18

Replace He emission lines and similar instances with HeI emission lines (same for Fe, which is typically FeII here).

Reply to the referee: This has now been fixed.

3.19 ✓ Comment 19

In all figures, make sure tick marks are present at bottom and top of plots. Use major as well as minor tick marks. It is hard to interpret figures without these.

Reply to the referee: Added tick marks to the top of plots where missing.

3.20 ✓ Comment 20

Times are confusing (even if the nomenclature is defined early on). The authors use +N days for N days post discovery but this usually refers to days post bolometric or B-band maximum. To avoid confusion, I would suggest always dropping the "+" and just state pre or post-discovery, or pre- or post-maximum etc.

Reply to the referee: We have removed the "+" signs and instead now mention if the phase is post first detection, or post maximum.

3.21 ✓ Comment 21

p27: quasi-continuum is awkward when referring to the emission blueward of 5700A and interpreted as arising from FeII lines rather than continuum emission. Some reference would be good too. Referring to FeII line emission (or blue quasi-featureless emission) would be more adequate.

Reply to the referee: Please note that this feature is commonly described as a quasi- or pseudo-continuum in the literature, we have now also added some of this literature in this section as reference.

3.22 ✓ Comment 22

”receding red wing” : awkward.

Reply to the referee: Rephrased.

4 Referee 3

4.1 ✓ Comment 1

24 – “from a companion star”, reference 1: I am not sure this is the appropriate reference, as the original suggestion came much earlier than 2011.

Reply to the referee: This is a fair point, we have changed the references to Whelan 1973 and Nomoto 1982 ((see page 1).

4.2 ✓ Comment 2

35-38 : Although it is understood that the space is limited, the authors should also briefly mention the search for circumstellar material placed at distances sufficiently large that no interaction (and hence no radio emission) is expected. The work done on the search for narrow absorption lines should be cited (Patat+ 2007, Simon+ 2007, Maguire+ 2013, Sternberg+ 2014). Incidentally, the first paper introduced the possibility that recurrent pre-explosion nova outbursts could excavate the CSM, hence weakening the link between lack of radio detections and DD progenitors).

Reply to the referee: Thanks for the suggestion. We now mention this in the CSM shells section, including the references (see page 31).

4.3 ✓ Comment 3

62: reference 30. This refers to a paper that deals with the same topic, in general terms. However, that article reports the detection of narrow absorption lines from neutral CSM, and has nothing to do with prominent Balmer emission lines. Although the paper should be referred to in the general introduction (see the comment above), the reference here is not appropriate.

Reply to the referee: Thanks for pointing this out, we moved the reference.

4.4 Comment 4

106: “Although the thermonuclear nature of Type Iax SNe is debated” : It would be useful to add a reference in support of this statement.

Reply to the referee: We have added a reference to the text.

4.5 ✓ Comment 5

150-151: Although interesting, this sentence is not substantiated by quantitative arguments. One would expect that the circumstellar disk is comparatively small, and hence it should be quickly overtaken by the SN explosion, which

would most likely also evaporate the dust. If that is the case, it is hard to identify this dust as the source of the IR emission from the SN. The authors should briefly discuss this.

Reply to the referee: The dust torus in V445 Puppis has an outer radius of $\gtrsim 10^{16}$ cm (Sect. 6). The dust sublimation radius of a SN Ia is of order 10^{16} - 10^{17} cm (Sect. 5), depending on peak luminosity and dust composition. So it is conceivable a shell of dust survives the SN Ia in a V445 Puppis scenario, in particular if the SN Ia is subluminous. We have added a sentence to Sect. 6 to this effect (see page 32).

4.6 ✓ Comment 6

167-170: The authors state that the He start+WD is likely the dominant source of short-delay SNIa. However, no reference is given in support of this statement. Also, there is no context for the reader to judge how important this contribution would be, for instance, in terms of rates.

Reply to the referee: We have added a reference to support this statement, as well as a rate estimate of SNe Ia arising from the He + WD formation channel (10% of all SD SNe Ia, Ruiter+09, see page 5)

4.7 ✓ Comment 7

176-177: The authors state that this is the first SN Ia where a SD origin is confirmed through a radio detection. The presence of a dense and detached CSM is witnessed by the light curve and the narrow emission lines in the spectra. The nature of the donor is deduced from the strength of He lines and the weakness of the H lines. Although the radio detection is certainly important, it is not clear why it should confirm the SD nature of the progenitor. This can probably be fixed by replacing “confirmed through” with “supported by”.

Reply to the referee: We did adopt the suggested phrasing, but in editing the main text this sentence was dropped altogether.

Reviewer Reports on the First Revision:

Referees' comments:

Referee #1 (Remarks to the Author):

The authors have nicely addressed my concerns in this re-write. I believe the wording in the associated figures highlights these points well now and allows the readers to better understand their conclusions.

Referee #2 (Remarks to the Author):

I am satisfied with the response and changes made to the manuscript and have no further comment. I therefore give my support for the publication in Nature of these unique observations and important implications for transient science.

Referee #3 (Remarks to the Author):

I am satisfied with the feedback and changes implemented by the authors. I do recommend the paper is published.